# Solving Robust MDPs through No-Regret Dynamics

**Etash Guha**                                                          *etash.guha@sambanovasystems*
*SambaNova Systems, University of Washington*

**Reviewed on OpenReview:** *https://openreview.net/forum?id=SdCuffxg5A*

## Abstract

Reinforcement learning is a powerful framework for training agents to navigate different situations, but it is susceptible to changes in environmental dynamics. Generating an algorithm that can find environmentally robust policies efficiently and handle different model parameterizations without imposing stringent assumptions on the uncertainty set of transitions is difficult due to the intricate interactions between policy and environment. In this paper, we address both of these issues with a No-Regret Dynamics framework that utilizes policy gradient methods and iteratively approximates the worst case environment during training, avoiding assumptions on the uncertainty set. Alongside a toolbox of nonconvex online learning algorithms, we demonstrate that our framework can achieve fast convergence rates for many different problem settings and relax assumptions on the uncertainty set of transitions.

## 1 Introduction

Reinforcement learning (RL) is a powerful subset of machine learning that enables agents to learn through trial-and-error interactions with their environment. RL has succeeded in various applications such as game playing, robotics, and finance (Sutton & Barto, 2018). However, when a trained policy operates in different environmental dynamics than the training environment, it often underperforms and achieves suboptimal rewards (Farebrother et al., 2018; Packer et al., 2018; Cobbe et al., 2018; Song et al., 2019; Raileanu & Fergus, 2021). Mitigating disastrous failures of RL-trained agents in practice can prevent many undesirable outcomes (Srinivasan et al., 2020; Choi et al., 2021). Robust Markov Decision Processes (MDPs) have emerged as a promising solution to mitigate this problem and address the issue of the sensitivity of reinforcement learning to changing environments. The Robust MDP problem is finding a policy that maximizes some chosen objective function in the worst-case environment. By optimizing policies against hostile environmental dynamics, Robust MDPs provide a more stable approach to training agents (Nilim & Ghaoui, 2003; Bagnell et al.; Iyengar, 2005).

While the Robust MDP is a powerful framework for changing environments, designing robust algorithms that solve such MDPs still presents significant challenges. One primary difficulty arises from the nonconvexity of many objective functions in Robust MDPs. Additionally, the intricate interactions between policy and environment in many MDPs make the problem difficult to handle. Many Value Iteration based approaches have been studied to solve Robust MDPs under a Direct Parameterization setting (Nilim & Ghaoui, 2003; Iyengar, 2005; Badrinath & Kalathil, 2021; Wiesemann et al., 2013; Roy et al., 2017; Tamar et al., 2014). However, many recent works have focussed on extending these methodologies past the direct parameterization setting (Dong et al., 2022; Wang & Zou, 2022) by utilizing modern policy gradient methods. Such gradient techniques are popular for their scalability and versatility to many different problem settings (Williams, 1992; Sutton et al., 1999; Konda & Tsitsiklis, 1999; Kakade, 2001). Most works make convergence explicit in Robust MDPs with such gradient techniques often by assuming that the set of possible adversarial environments, known as the uncertainty set, take on specific shapes, such as Rectangular sets (Dong et al., 2022), $R$-Contamination sets (Wang & Zou, 2022), or Wasserstein balls Clement & Kroer (2021). Finding algorithms that can solve Robust MDPs for general shapes of uncertainty sets is a complex and open problem.

To more fundamentally attack this problem, we want to form an algorithmic framework to design algorithms to solve Robust MDPs under many different settings. Several works have solved finding robust minima in different problem settings using a No-Regret Dynamics framework (Wang et al., 2022a; Balduzzi et al., 2018; Syrgkanis et al., 2015). Such a framework frames an algorithm to find robust minima as a two-player game between a learner and the environment. These frameworks enjoy both versatility in design and simplicity of analysis. Most importantly, the convergence of an algorithm from this framework can be bounded in terms of the expected regrets of both players' online strategies without strong assumptions. We use a similar framework style to attack the Robust MDP to more fundamentally attack the challenges of limiting assumptions and versatility to different settings. Our proposed game precisely consists of two players: policy and environment players. In each round of the game, the policy player first picks a policy and tries to maximize its objective function. After that, the environment observes the policy and then chooses a hostile environment to minimize the policy player's objective function. The two players repeatedly play against each other, and the goal is to converge to a robust minimum, which naturally yields a policy that is robust to worst-case environments. This framework has three concrete advantages. First, in each round, the framework calls for explicitly approximating the worst-case environment. This step is critical to relax assumptions on the shape of uncertainty set of environments. Second, both players can choose any online strategy, including gradient-based updates. This flexibility makes this framework applicable to a host of problem settings. Third, the time taken till the algorithm converges to a robust equilibrium is bounded by the regret of the two players' chosen strategies. This makes both algorithm design and convergence analysis simple.

To utilize our framework, we provide various possible online strategies for each player. Given the nonconvexity of the objective functions in Robust MDPs, we explore the existing nonconvex online learning literature to develop our toolbox and augment it with several new nonconvex online algorithms that work well in our framework. For the first augmentation, we make use of the fact that the environment player in our framework can see the incoming loss function and developed two new algorithms known as Best-Response and Follow the Perturbed Leader Plus (FTPL+), which enjoy improved regret rates with this ability to peak at the policy player's chosen policy. The second augmentation is that the nonconvex online learning literature often depends on a minimization oracle that can find a global minimizer of the nonconvex loss function (Suggala & Netrapalli, 2019). While this is impossible to build in every setting, we identify the MDP settings under which such an oracle is accessible. To be more specific, we demonstrate that for both the policy and environmental players, the Value Function, a common objective for Robust MDPs, exhibits *gradient-dominance*. Thus, using Projected Gradient Descent (PGD) will surprisingly suffice as a minimization oracle for both players. This choice has the added benefit of enjoying the scalability of gradient methods. Thus, under different conditions, including simple gradient-dominance, smooth MDPs, or strongly gradient-dominated MDPs, we build different algorithms from our framework using different algorithms that take advantage of each setting. With our framework, proving convergence for each algorithm is simple yet powerful. For example, in the most common setting where the objective function is the Value Function, our algorithm achieves the strong convergence rate of $\mathcal{O}\left(T^{-\frac{1}{2}}\right)$ where $T$ is the number of iterations of our algorithm. This rate is competitive with the most recent Robust MDP algorithms from the literature. Moreover, due to the explicit approximation of the worst-case environment throughout the training process, we do not need an assumption on the uncertainty set shape; instead, we only need the uncertainty set to be convex.

We utilize our algorithmic framework to provide different algorithms for Robust MDPs with gradient-dominance, smooth MDPs, and strongly gradient-dominated MDPs. We also prove robust convergence rates with only a convexity assumption on the uncertainty set of environments. Alongside these guarantees, we run several small experiments on the convergence behavior of our algorithm where the Value Function is the optimization objective in the GridWorld MDP. Our small experiments corroborate this convergence rate in Section 7.

**Contributions**   Our contributions are as follows.

1. We design an algorithmic framework for generating algorithms to solve Robust MDPs based on No-Regret Dynamics. Alongside our framework, we develop novel, no-regret online learning algorithms to use with our framework. This framework is versatile and can work for various problem settings, including direct parameterization. Moreover, due to the explicit approximation of the worst-case

environment during training, we need only the uncertainty set of transitions to be convex to guarantee convergence.

2. Many of the no-regret online algorithms in our framework require access to a Minimization Oracle. We show that when the objective function of the game is gradient-dominated for both players, we can use Projected Gradient Descent as the oracle. We also provide tools for identifying the conditions under which the gradient-dominated property of an objective function is ensured. For example, we show that when the MDPs' objective function is the Value Function in the tabular setting, both players' objective functions enjoy gradient-dominance. We then prove that when the objective function is the Value Function in the direct parameterization setting, our framework can generate an algorithm that yields an $\mathcal{O}\left(T^{-\frac{1}{2}}\right)$ convergence rate. We empirically verify that our algorithm finds a robust policy at a rate of roughly $\mathcal{O}\left(T^{-\frac{1}{2}}\right)$ in the GridWorld MDP across several uncertainty set shapes and sizes.

3. Finally, we demonstrate that even faster convergence rates can be obtained when the objective function is Lipschitz smooth or enjoys strong gradient-dominance with advanced online dynamics. To our knowledge, this is the first work to study Robust MDPs with strong gradient-dominance.

## 2 Related Works

**Robust MDPs**  Some of the first algorithms to solve Robust MDPs with convergence guarantees and empirical performance are via transition dynamics set assumptions(Nilim & Ghaoui, 2003; Iyengar, 2005), Robust Policy Iteration (Mankowitz et al., 2019; Tamar et al., 2014; Sinha & Ghate, 2016) or Robust Q-Learning (Wang & Zou, 2021). After the introduction of Robust MDPs, several works studied the setting where only samples from the MDP are accessible (Wang & Zou, 2022; Badrinath & Kalathil, 2021; Roy et al., 2017; Tessler et al., 2019). Goyal & Grand-Clement (2023) expanded the rectangular uncertainty set using Factor Matrix Uncertainty Sets to be more general. Zhang et al. (2021) introduced a different adversary where the stochastic samples from the central MDP are adversarial. This work focuses solely on optimizing the policy where the uncertainty set is known. Regarding robust policy optimization, Mankowitz et al. (2018) proposed robust learning through policy gradient for learning temporally extended action. Mankowitz et al. (2019) used Bellman contractors to optimize the robust objective but did not provide any sub-optimality guarantees, only convergence. Dong et al. (2022) used an online approach similar to ours but used rectangular uncertainty set assumptions and did not utilize any no-regret dynamics. Wang et al. (2023) and Tamar et al. (2014) discussed a policy iteration approach to improving the Average Reward Robust MDP using Bellman Equations. Wang et al. (2022b) uses a similar game framework but does not use no-regret dynamics or sophisticated online learning algorithms to achieve their guarantees, achieving a worse convergence rate with less robust guarantees.

Wang & Zou (2022) is one of the best-known algorithms for solving Robust MDPs in this setting with policy gradient methods. Unlike our work, they rely on the uncertainty set taking the form of $R$-Contamination set, while our algorithm needs no such assumption. Moreover, Clement & Kroer (2021) do approximate the worst-case environment iteratively. However, unlike our work, they utilize Wasserstein uncertainty set assumptions and focus on direct parameterizations. There are also other forms of Robust MDPs, including distributionally robust MDPs (Xu & Mannor, 2010; Ruszczyński, 2010; Shapiro, 2016; Xu & Mannor, 2009; Petrik, 2012; Ghavamzadeh et al., 2016; Chen et al., 2019), soft-robust MDPs (Buchholz & Scheftelowitsch, 2019; Lobo et al., 2020; Steimle et al., 2021), and noise robustness (Pattanaik et al., 2017; Eysenbach & Levine, 2021).

**No-Regret Learning**  In the context of game no-regret learning, McMahan & Abernethy (2013) used a similar game framework to solve linear optimization. In contrast, Wang et al. (2022a) used this framework for solving binary linear classification. Wang et al. (2021) used these frameworks to solve Fenchel games, phrasing many optimization algorithms. Daskalakis et al. (2017) showed that using a similar framework using online players for training GANs yields strong theoretical and empirical improvements. The classical Syrgkanis et al. (2015) uses No-Regret Dynamics to solve social welfare games. Balduzzi et al. (2018) helped show the convergence of these games even in adversarial settings and explained gradient descent as a two-player game.

## 3    Preliminaries

In this section, we present the problem setup along with some standard assumptions and definitions.

### 3.1    Notation

We denote our infinite horizon, $\gamma$-discounted MDP as a tuple of $(\mathcal{S}, \mathcal{A}, \mathbb{P}_W, R)$, where $\mathcal{S}$ is a set of states, $\mathcal{A}$ is a set of actions, $\mathbb{P}_W$ is a transition dynamics function parameterized by $W$ that maps a state-action-state triple $s, a, s' \in \mathcal{S} \times \mathcal{A} \times \mathcal{S}$ to a probability between 0 and 1, and $R$ is a reward function that maps a state $s$ to a reward $r$. Here, $W$ parameterizes the transition dynamics belonging to some bounded convex set $\mathcal{W}$. Moreover, we will denote $R_{\max}$ as a constant denoting the largest possible value of $R$. We will assume that both $\mathcal{S}$ and $\mathcal{A}$ are finite sets. We will design a policy $\pi_\theta$ parameterized by a term $\theta \in \mathcal{T}$ where $\mathcal{T}$ is a bounded convex set of vectors of dimension $d$ for some constant $d$. This policy $\pi_\theta$ maps a state action pair $s, a \in \mathcal{S} \times \mathcal{A}$ to a probability $[0, 1]$. For most of this paper, we will refer to $\pi_\theta$ as $\pi$ where clear. We will not specify the form $W$ and $\theta$ take since that depends on the parameterization of the policy and transition dynamics. For example, under direct parameterization for both transition dynamics and policy, $W$ belongs to probability simplex $W \in \Delta^{|\mathcal{S}| \times |\mathcal{A}| \times |\mathcal{S}|}$ and $\theta$ belongs to the probability simplex $\theta \in \Delta^{|\mathcal{S}| \times |\mathcal{A}|}$.

We denote the value function $V$ of a state $s$ under the policy $\pi$ and transition dynamics $\mathbb{P}_W$ as $V_W^\pi(s)$. This function is the expected value function of arriving in a state $s$. We will define $\mu \in \Delta^{|\mathcal{S}|}$ as some probability distribution over the initial states. Moreover, we will slightly abuse notation and call $V_W^\pi(\mu) = \mu^\top V_W^\pi$ where $V_W^\pi$ is the vector of value functions for all states. Moreover, we will define $d_{s_0}^W(s)$ and $d_{s_0}^\pi(s)$ as the probability distribution over the occupancy of states when a policy $\pi$ interacts with an environment of dynamics $W$ given the initial state $s_0$. Formally, this is written as

$$d_{s_0}^W(s) = d_{s_0}^\pi(s) = (1 - \gamma) \sum_{t=0}^\infty \gamma^t \mathbb{P}(s_t = s | s_0, \pi, W).$$

We choose to use either $d_{s_0}^W(s)$ and $d_{s_0}^\pi(s)$ based on the context. Moroever, the state visitation distribution under initial state distribution $\mu$ is formally $d_\mu^\pi(s) = \mathop{\mathbb{E}}_{s_0 \sim \mu}[d_{s_0}^\pi(s)]$ and $d_\mu^W(s) = \mathop{\mathbb{E}}_{s_0 \sim \mu}[d_{s_0}^W(s)]$.

### 3.2    Robust Policies

Our main goal is to create a policy $\pi$ where some objective function over the initial state distribution is as large as possible against the worst environments. Within different formulations of Robust MDPs, there may be different objective functions for the robust optimization, which we denote as $g(\pi, W)$. One common setup of Robust MDPs is where $g(\pi, W)$ is simply the Value Function given $\pi$ and $W$, specifically $g(\pi, W) = V_W^\pi(\mu)$.

**Definition 3.1.** *We are in the infinite horizon setting where $\gamma$ serves as a discount factor and is a positive constant less than 1. Given a policy $\pi$ and a transition dynamics $\mathbb{P}_W$, we define the value function recursively as*

$$V_W^\pi(s) = R(s) + \gamma \sum_{a \in \mathcal{A}} \pi(a, s) \sum_{s' \in \mathcal{S}} \mathbb{P}_W(s', a, s) V_W^\pi(s').$$

In this setting, the Robust MDP problem is simply the task of finding a policy $\pi$ that maximizes the Value Function under worst-case transition dynamics as in

$$\pi = \arg\max_{\pi \in \mathcal{T}} \min_{W \in \mathcal{W}} g(\pi, W) = \arg\max_{\pi \in \mathcal{T}} \min_{W \in \mathcal{W}} V_W^\pi(\mu).$$

However, in different setups such as Control or Regularized MDPs (Bhandari & Russo, 2022), it may be desirable to find a policy solving a similar robust optimization problem with a different objective function. For example, one may wish to regularize the policy parameter as in $g(\pi_\theta, W) = V_W^{\pi_\theta}(\mu) - \|\theta\|^2$. To generalize the Robust MDP problem to all such possible MDPs, we will denote the optimization problem as the following.

**Definition 3.2.** *The Robust MDP problem is that of finding $\pi$ such that $\pi = \arg\max_{\pi \in \mathcal{T}} \min_{W \in \mathcal{W}} g(\pi, W)$.*

To connect this more clearly to work on repeated games, we will view this as finding a policy $\pi$ that minimizes the suboptimality, i.e., $\arg\max_{\pi^* \in \mathcal{T}} \min_{W \in \mathcal{W}} g(\pi^*, W) - \min_{W \in \mathcal{W}} g(\pi, W)$. Improving the robustness of the policy can similarly be seen as reducing the difference between our policy's robustness and the best policy's robustness. The second definition intuitively connects to the online learning literature definition of regret. We will similarly use this algorithmic perspective to design a Robust Policy Optimization algorithm that minimizes the suboptimality of the learned policy with the least amount of computational complexity possible. Similar min-max games have been solved in the past using No-Regret Dynamics, a powerful and versatile framework for algorithm design. Given its existing connection to robustness (Wang et al., 2022a), we explore how to use a No-Regret Dynamics based framework to solve the Robust MDP problem.

## 4   No Regret Dynamics

In this section, we present a general No-Regret Dynamics framework. We then demonstrate how such a framework can be used to generate algorithms for Robust MDPs.

### 4.1   The General Framework

Our No-Regret framework is presented in Algorithm 1. Intuitively, we iteratively choose a policy that performs well in previously seen adversarial environments, and we iteratively look for adversarial environments. In the end, the policies produced at the later rounds will become more and more robust. Specifically, in each round $t$ of this algorithm, our policy player called $\text{OL}^\pi$, chooses a policy $\pi_t$ at time step $t$. Our second player is a $W$-player, called $\text{OL}^W$,

---
**Algorithm 1:** No-Regret RL
---
**Data:** $T$

**for** $t \in [T]$ **do**
    $\text{OL}^\pi$ **chooses a policy:** $\pi_t \leftarrow OL^\pi$;
    $\text{OL}^W$ **sees the policy:** $OL^W \leftarrow \pi_t$;
    $\text{OL}^W$ **chooses transition dynamics:** $W_t \leftarrow OL^W$;
    $\text{OL}^\pi$ **sees the transition dynamics:** $OL^\pi \leftarrow W_t$;
    $\text{OL}^W$ **incur losses:** $OL^W \leftarrow l_t(W_t)$;
    $\text{OL}^\pi$ **incur losses:** $OL^\pi \leftarrow h_t(\pi_t)$;
**end**

---

which will see the policy $\pi_t$ outputted by the policy player and, then, outputs a transition dynamic $W_t$. The policy player then sees the $W_t$ that was chosen. The policy player incurs a loss $h_t(\pi_t)$, and the environment player then incurs a loss $l_t(W_t)$ corresponding to their decision. They repeatedly play this game until an equilibrium is reached. Here, the online players $\text{OL}^\pi$ and $\text{OL}^W$ can be any online strategy. In Wang et al. (2022a), example strategies for online players include Mirror Descent and Follow the Leader. The performance of a strategy for either online players can be measured in terms of regret. By definition, the regret of the $W$-player is equivalent to

$$\text{Reg}_W = \sum_{t=0}^{T} l_t(W_t) - \min_{W^*} \sum_{t=0}^{T} l_t(W^*)$$

Similarly, for the $\pi$ player, we have that

$$\text{Reg}_\pi = \sum_{t=0}^{T} h_t(\pi_t) - \min_{\pi^*} \sum_{t=0}^{T} h_t(\pi^*)$$

This framework is a simple and versatile method of solving many optimization problems. We need only choose the online algorithms that the $\pi$-player and $W$-player employ and the loss function they see. The benefit of a No-Regret Dynamics framework is that when the loss functions for both players are negative of the others, the algorithm's convergence is simply the convergence for two players. By setting $l_t(W_t) = g(W_t, \pi_t)$ and $h_t(\pi_t) = -g(W_t, \pi_t)$, we have Theorem 4.1.

**Theorem 4.1.** *We have the difference between the robustnesses of the chosen policies and any policy $\bar{\pi}$ is upper bounded by the regret of the two players*

$$\frac{1}{T} \min_{W^* \in \mathcal{W}} \sum_{t=0}^{T} g(W^*, \bar{\pi}) - \frac{1}{T} \min_{W^* \in \mathcal{W}} \sum_{t=0}^{T} g(W^*, \pi_t) \leq Reg_W + Reg_\pi.$$

*Here, $Reg_W$ and $Reg_\pi$ are the two average regrets of the two players $OL^W$ and $OL^\pi$.*

As in Theorem 4.1, the convergence of Algorithm 1 is explained by the regret of the two players. This framework is both powerful and intuitive. Say we are in the traditional Robust MDP setting where $g(W, \pi) = V_W^\pi(\mu)$. We can get convergence guarantees for the robustness of our setting by configuring the loss functions $l_t$ and $h_t$ in the following way. In this setting, the $t$th loss function for the $W$-player is $l_t(W_t) = V_{W_t}^{\pi_t}(\mu)$ and for the $\pi$-player is $h_t(\pi_t) = -V_{W_t}^{\pi_t}(\mu)$. Setting the loss functions according to the above, we similarly get the following convergence guarantee based on the regret of the two players, just like in Theorem 4.1. Therefore, despite the nonconvexity, we have a simple framework for solving the Robust MDP problem. Since we approximate the worst-case environment at every round, we only need an assumption that the uncertainty set of transitions is convex. Moreover, we have the flexibility to generate many algorithms for different settings by choosing different online learning algorithms for both players. However, many recent online learning results require the underlying loss functions to be convex or strongly convex. Regrettably, in the simplest setting where $g(W, \pi) = V_W^\pi(\mu)$, neither of these properties are satisfied. Therefore, we must look for efficient online learning algorithms for nonconvex loss functions. We now present a toolbox of such online learning algorithms.

## 4.2   Online Learning Toolbox

To generate an algorithm from our No-Regret framework in Algorithm 1, we need only choose strategies for both players. Here, we develop a toolbox of nonconvex online learning algorithms usable in our framework. We summarize our toolbox in Algorithm 2.

**Online algorithms for the $\pi$-player**   First, we present two existing algorithms from Suggala & Netrapalli (2019), namely Follow the Perturbed Leader and Optimistic Follow the Perturbed Leader, which achieves strong convergence in expectation in nonconvex settings and is suitable for the $\pi$-player. At each time step, the Follow the Perturbed Leader algorithm generates a random noise vector $\sigma_t$ according to an Exponential distribution with parameter $\eta$. Intuitively, this noise vector helps the online learner avoid local minima. It then chooses $x_t = \mathcal{O}_\alpha \left( \sum_{i=1}^{t-1} h_i - \sigma_i \right)$. Here, $\mathcal{O}_\alpha$ is an Approximate Optimization Oracle that approximately minimizes the received loss function $h_i$ to accuracy $\alpha$. The Optimistic Follow the Perturbed Leader follows a similar procedure, except it chooses $x_t = \mathcal{O}_\alpha \left( m_t + \sum_{i=1}^{t-1} h_i - \sigma_i \right)$ where $m_t$ is some optimistic function. The main dependency of their regret bounds is that there exists an Approximate Optimization Oracle $\mathcal{O}_\alpha$.

**Definition 4.1.** *An optimization oracle is a function that takes some nonconvex function $f$ and returns an approximate minimizer $x^*$ such that*

$$f(x^*) - \langle \sigma, x^* \rangle \leq \left[ \inf_x f(x) - \langle \sigma, x \rangle \right] + \alpha.$$

Throughout this section, we will assume the presence of such an oracle and discuss how to set this oracle later. Suggala & Netrapalli (2019) prove a regret bound for Follow the Perturbed Leader and Optimistic Follow the Perturbed Leader algorithms when they can access such an oracle.

**Online algorithms for the $W$-player**   Note that, for our framework, the environmental player can see the incoming loss function before choosing an environment, but neither of the aforementioned methods can account for this. To address this problem, we prove that a "Best Response" algorithm achieves an even smaller upper bound regarding regret. The Best Response algorithm assumes knowledge of the coming loss function and returns the value that minimizes the incoming loss function. Formally, the Best Response algorithm outputs $x_t = \arg\min_x l_t(x)$. This is a useful algorithm given that the $W$ player can see the output of the $\pi$ player when making its decision and greatly reduce its regret. We prove this in the following regret bound.

**Lemma 4.1.** *Suppose we have an incoming sequence of loss functions $f_t$ for $t \in [T]$ with an optimization oracle that can minimize a function to less than $\alpha$ error. The Best Response algorithm satisfies the following regret bound*

$$\frac{1}{T} \sum_{t=1}^{T} f_t(x_t) - \frac{1}{T} \inf_{x \in \mathcal{X}} \sum_{t=1}^{T} f_t(x) \leq \alpha.$$

---

**Algorithm 2:** A set of useful online learning algorithms

---

**Input:** $\eta, \mathcal{O}_\alpha$

**for** $t = 1, \dots, T$ **do**

$$\text{Sample } \sigma \in \mathbb{R}^d \text{ and } \sigma_i \sim \text{Exp}(\eta) \text{ where } i \in [d]$$

$$\text{FTPL} : x_t = \mathcal{O}_\alpha \left( \sum_{i=1}^{t-1} f_i(x) - \langle \sigma, x \rangle \right)$$

$$\text{OFTPL}[m_t] : x_t = \mathcal{O}_\alpha \left( \sum_{i=1}^{t-1} f_i(x) + m_t(x) - \langle \sigma, x \rangle \right)$$

$$\text{Best-Response} : x_t = \mathcal{O}_\alpha \left( f_t(x) \right)$$

$$\text{FTPL+} : x_t = \mathcal{O}_\alpha \left( \sum_{i=1}^{t} f_i(x) - \langle \sigma, x \rangle \right)$$

**end**

---

We will also prove the regret of one more algorithm, Follow the Perturbed Leader Plus. Similar to the Follow the Leader Plus algorithm from Wang et al. (2021), Follow the Perturbed Leader Plus assumes knowledge of the incoming loss function and then outputs the minimizer of the sum of all the seen loss functions minus the noise term. Formally, Follow the Perturbed Leader Plus outputs $x_t$ that satisfies $x_t = \arg\min_{x_t} \sum_{i=1}^{t} l_i(x_t) - \langle \sigma, x_t \rangle$.

While this algorithm achieves worse regret than Best Response, it produces more stable outcomes. This will be useful for later extensions, such as Smooth Robust MDPs. In fact, Follow the Perturbed Leader Plus is equivalent to OFTPL, when the optimistic function $m_t = l_t$ is the true loss function $l_t$. We present the regret of this algorithm here.

**Lemma 4.2.** *Let $\eta$ be the constant parameterizing the exponential distribution from which the noise $\sigma_t$ is drawn and $d$ be the dimensionality of the set of choices $\mathcal{X}$. Moreover, denote $D$ as the maximum distance between any two points in $\mathcal{X}$, i.e. $D = \max_{x,x' \in \mathcal{X}} \|x - x'\|_2$. Assume access to an optimization oracle that yields solutions with at most error $\alpha$. Given a series of choices by FTPL+ $x_1, \dots, x_T$ with loss functions $l_1, \dots, l_T$, the regret in expectation is upper bounded by*

$$\mathbb{E}\left[ \frac{1}{T} \sum_{t=1}^{T} l_t(x_t) - \frac{1}{T} \inf_{x \in \mathcal{X}} \sum_{t=1}^{T} l_t(x) \right] \le O\left( \frac{dD}{\eta T} + \alpha \right).$$

We summarize these online strategies in Algorithm 2. These two algorithms are suitable choices for the environmental player, and the two algorithms from Suggala & Netrapalli (2019) are suitable choices for the policy player. However, we still have the issue that an Approximation Optimization Oracle is generally challenging to parameterize. However, we exploit a unique characteristic of many Robust MDP variants. While the objective functions of Robust MDPs do not exhibit convexity, they exhibit *Gradient-Dominance* in some settings. This property helps us design a sufficient Approximation Oracle.

## 5    Constructing the Optimization Oracle

The online learning algorithms demonstrated in the previous section require *having access to an approximate optimization oracle*. However, designing the approximate optimization oracle can be a complex problem. In this section, we first show that such an oracle can be easily constructed when the objective function is gradient-dominated. We then demonstrate that under direct parameterization, the gradient-dominance property is satisfied.

### 5.1 Gradient-Dominance

A function is gradient-dominated if the difference between the function value at a point $x$ and the minimal function value is upper bounded on the order of the function's gradient at the point $x$. We formalize this in the below definition.

---

**Algorithm 3:** Projected Gradient Descent

**Data:** Initial guess $x_0$, step size $\beta$, projection operator $\text{Proj}_{\mathcal{X}}$

**for** $t = 0$ *to* $T_{\mathcal{O}} - 1$ **do**
$\quad | \quad x_{t+1} \leftarrow \text{Proj}_{\mathcal{X}}(x_t - \beta\nabla f(x_t))$
**end**

---

**Definition 5.1.** *We say a function $f$ is gradient-dominated for set $\mathcal{X}$ with constant $\mathcal{K}$ if*

$$f(x) - \min_{x* \in \mathcal{X}} f(x^*) \leq -\mathcal{K}\min_{\bar{x} \in \mathcal{X}}\langle \bar{x} - x, \nabla f(x)\rangle.$$

*Here, $\mathcal{K}$ is some constant greater than $0$.*

As noted by Bhandari & Russo (2022), this gradient-dominated property is relatively common for many different RL settings, including Quadratic Control or direct parameterization. It is useful for proving convergence guarantees for traditional policy gradient methods (Agarwal et al., 2019). For gradient-dominated functions, one can use projected gradient descent to minimize the function $f$. We recap projected gradient descent in Algorithm 3. Namely, Bhandari & Russo (2022) shows that

**Lemma 5.1.** *The below property only holds if $f$ is gradient-dominated and $\beta \leq \min\left\{\frac{1}{\sup\limits_{x}\|\nabla f(x)\|_2}, \frac{1}{L}\right\}$. Here, $T_{\mathcal{O}}$ is the number of iterations of Projected Gradient Descent runs. Moreover, the function $c_x$ is used for brevity for $c_{\pi}$ and $c_W$ later. Also, $D = \max\limits_{x,x' \in \mathcal{X}}\|x - x'\|_2$. Given that $\nabla f$ is L-Lipschitz continuous, the sequence $x_{t+1} = \text{Proj}_{\mathcal{X}}(x_t - \beta\nabla f(x_t))$ enjoys the property*

$$f(x_{T_{\mathcal{O}}}) - \inf_{x^*} f(x^*) \leq \sqrt{\frac{2D^2\mathcal{K}^2(f(x_0) - \inf\limits_{x^*} f(x^*))}{\beta T_{\mathcal{O}}}}$$
$$= c_x(T_{\mathcal{O}}, \mathcal{K}).$$

Ideally, we would like to use projected gradient descent as our Approximate Optimization Oracle for $OL^{\pi}$ and $OL^W$ since it is simple and utilizes scalable gradient-based techniques. However, this would require the loss functions $l_t$ and $h_t$ to be gradient-dominated. While not generally true, this is known to hold in many cases. Gradient-dominance is a well-known phenomenon for the $\pi$-player when the Value Function is the objective function, as seen in Agarwal et al. (2019). We formally list some helpful conditions here.

**Condition 5.1.** *Here, we list the conditions we have.*

1. *The function $\sum_i^t f_i(x) - \langle\sigma, x\rangle$ is gradient-dominated, enabling the use of FTPL+*

2. *The function $\sum_i^{t-1} f_i(x) - \langle\sigma, x\rangle$ is gradient-dominated, enabling the use of FTPL.*

3. *The function $\sum_i^{t-1} f_i(x) + f_{t-1}(x) - \langle\sigma, x\rangle$ is gradient-dominated, enabling the use of OFTPL.*

4. *The function $f_t(x)$ is gradient-dominated, enabling the use of Best Response.*

We will first provide tools useful for showing when these conditions hold.

### 5.2 Tools for Demonstrating Gradient-Dominance

Here, we will provide the tools to show that any of these conditions hold for the objective functions for either the $\pi$ or $W$ players. We have a gradient term within the terms of gradient-dominance from Definition 5.1. In many cases, the loss function will often contain the Value Function. Therefore, we must know what the gradients of the Value Functions will be for both players. While this is known for the policy player from the Policy Gradient Theorem (Sutton & Barto, 2018), we demonstrate a similar result for the gradient of the Value Function for the $W$-player.

**Lemma 5.2.** *The gradient of the value function $V^W$ with respect to the parameter $W$, denoted as $\nabla_W V^W(s)$, is given by*

$$\nabla_W V^W(s) = \frac{1}{1-\gamma} \sum_{s',a,s} d_\mu^W(s)\pi(a,s)\nabla\mathbb{P}_W(s',a,s)V^W(s').$$

Now, we express the suboptimality of a transition dynamics parameter $W$ in terms of the gradient of the Value Function.

While this is shown via the Performance Difference Lemma from Kakade & Langford (2002) for the policy player, we need a similar lemma for the transition dynamics. The Performance Difference Lemma relies on the *Advantage function*. We will define an analogous advantage function for the transition dynamics. Intuitively, this Advantage Function is the value of taking state $s'$ over the expected value over all states. Given this, we can provide an analog of the Performance Difference Lemma for the $W$-Player. We provide such a lemma here.

**Lemma 5.3.** *Given two different transition dynamics parameters $W$ and $W'$, we have that*

$$V_W(\mu) - V_{W'}(\mu) = \sum_{s',a,s} d_\mu^W(s)\mathbb{P}_W(s',a,s)\pi(a,s)A^{W'}(s',a,s).$$

*Here, we define the $W$-Advantage Function as $A^W(s',a,s) = \gamma V_W(s') + r(s,a) - V_W(s)$.*

We have presented the tools necessary to demonstrate gradient-dominance in many cases. While many MDP settings exhibit gradient-dominance (Bhandari & Russo, 2022), we demonstrate how to prove gradient-dominance for both players' loss functions under direct parameterization.

### 5.3 Direct Parameterization

We now begin with the direct parameterization case with standard Robust MDPs. This setting is one of the most standard settings for MDPs. Here, $\mathbb{P}_W(s',a,s) = W_{s',a,s}$ directly parameterizes the transition dynamics, $\pi(a,s) = \theta_{s,a}$, and $g(W,\pi) = V_W^\pi(\mu)$. Moreover, the set of transition dynamics is some convex bounded set, such as a rectangular uncertainty set (Dong et al., 2022). We demonstrate that in this setting, Subcondition 2 holds for the loss function of the policy player, and Subcondition 4 holds for the loss function of the $W$-player.

**Lemma 5.4.** *Let the objective function be the Value Function. For any positive noise term $\sigma$, we have that under direct parameterization, $\sum_i^{t-1} l_i(\cdot) - \langle\sigma,\cdot\rangle$ and $\sum_i^{t-1} h_i(\cdot) - \langle\sigma,\cdot\rangle$ are both gradient-dominated for the $W$-player and the $\pi$-player on sets $\mathcal{W}$ and $\mathcal{T}$ with constants $\mathcal{K}_W = \frac{1}{1-\gamma}\left\|\frac{d_\mu^{W^*}}{\mu}\right\|_\infty$ and $\mathcal{K}_\pi = \frac{1}{1-\gamma}\left\|\frac{d_\mu^{\pi^*}}{\mu}\right\|_\infty$ respectively. Therefore, Subcondition 2 hold for both the loss functions for both players.*

---

**Algorithm 4:** Algorithm under Direct Parameterization

**Data:** $T$

**for** $t \in [T]$ **do**

    Sample $\sigma_t \sim \mathrm{Exp}(\eta)$;

    $\pi_t \leftarrow \mathcal{O}_\alpha\left(\sum_{i=1}^{t-1} -V_{W_i}^{\pi_t}(\mu) - \langle\sigma_t, \pi_t\rangle\right)$ /* Follow the Perturbed Leader                */

    $W_t \leftarrow \mathcal{O}_\alpha\left(V_{W_t}^{\pi_t}(\mu)\right)$ /* Best-Response                                             */

**end**

---

Now, we have shown that when the objective function is the loss function, the loss functions for the policy player satisfy Subcondition 2. Therefore, we can use Follow the Perturbed Leader for $OL^\pi$. Now, we wish to show Subcondition 4 holds for the loss function of the $W$ player.

**Lemma 5.5.** *Under direct parameterization, we have that the $V_W(\mu)$ is gradient-dominated with constant $\mathcal{K}_W = \frac{1}{1-\gamma}\left\|\frac{d_\mu^{W^*}}{\mu}\right\|_\infty$ as in for an arbitrary $W \in \mathcal{W}$ and the optimal parameter $W^* \in \mathcal{W}$, we have*

$$V_W(\mu) - V_{W^*}(\mu) \leq -\mathcal{K}_W \min_{\bar{W} \in \mathcal{W}}\left[\left(\bar{W} - W\right)^\top \nabla_W V_W(\mu)\right].$$

*Here, we have that Subcondition 4 holds for the W-player.*

Now that we have that Subcondition 4 holds for the $W$ player, we know we can use Best Response for the $OL^W$ player. Thus, in the direct parameterization setting with standard Robust MDPs, we can use our framework from Algorithm 1 with Follow the Perturbed Leader for $OL^\pi$ and Best Response for $OL^W$ where both use Projected Gradient Descent as their Approximate Optimization Oracle. We present our algorithm in Algorithm 4. We can now prove the convergence and robustness of our framework using our proof framework.

### 5.4 Overall Convergence Rates

Now, if the policy player employs FTPL and the environment player employs Best-Response, we get the following convergence bound when the objective function is the Value Function.

**Theorem 5.1.** *Assume that the set of possible transition matrices $\mathcal{W}$ is convex. Let $L_\pi$ be the smoothness constant of $h_t$ with respect to the $\ell_1$ norm, $D_\pi = \max_{\pi,\pi' \in \mathcal{T}} \|\pi - \pi'\|_2$, and $d_\pi$ be the dimension of the input for the $\pi$-player. Let $OL^\pi$ use FTPL and $OL^W$ use Best-Response. Under direct parameterization, when $\eta = \frac{1}{L_\pi \sqrt{T d_\pi}}$ the robustness of the set of trained algorithms is for any $\bar{\pi}$*

$$\min_{W^* \in \mathcal{W}} \sum_{t=0}^{T} V_{W^*}^{\bar{\pi}}(\mu) - \mathbb{E}\left[\min_{W^* \in \mathcal{W}} \sum_{t=0}^{T} V_{W^*}^{\pi_t}(\mu)\right] \leq \frac{2 d_\pi^{\frac{3}{2}} D_\pi L_\pi}{\sqrt{T}} + c_\pi\left(T_\mathcal{O}, \mathcal{K}_\pi\right) + c_W\left(T_\mathcal{O}, \mathcal{K}_W\right).$$

We formalize this in Algorithm 4. Here, we have shown that in this simple setting, we have that the robustness in expectation is $\mathcal{O}\left(T^{-\frac{1}{2}} + T_\mathcal{O}^{-\frac{1}{2}}\right)$ with simple gradient-dominance assumptions.

### 5.5 Comparison to Existing Rates and Algorithms

Our resulting algorithm is similar to that of Wang et al. (2022b). Both algorithms iteratively optimize the policy and transition matrix in turns. However, their policy update is a single step of Projected Gradient Descent, and our framework utilized different updates for the policy player. With our framework, our method gets a more powerful convergence rate with less stringent assumptions. Moreover, Wang & Zou (2022) only update the policy on each step by using the gradient of the worst-case Value Function. However, their method relies on stringent assumptions on the uncertainty set to make the gradient computable. Clement & Kroer (2021) in structure is similar to that of Wang et al. (2022b) given that it iteratively updates both the policy and environment by directly optimizing the objective function. Instead, our method uses different updates, such as OFTPL. Overall, our algorithm differs from existing algorithms by optimizing both the policy and environment and exploring different updates for each, resulting in better convergence rates with weaker assumptions.

Both our algorithm and Wang & Zou (2022) achieve rates of roughly $\mathcal{O}\left(T^{-\frac{1}{2}}\right)$. However, they assume the uncertainty set is a $R$-Contamination set. Moreover, Clement & Kroer (2021) achieves a slightly faster rate of $\mathcal{O}\left(T^{-\frac{2}{3}}\right)$. However, their analysis is limited to the tabular setting and relies on the uncertainty set being a Wasserstein ball. To our knowledge, these are the fastest rates in the literature that solve Robust MDPs with gradient-based updates. Thus, our work roughly meets or comes near the fastest rates for solving Robust MDPs while requiring the least stringent assumptions. However, given some additional properties, it may be possible to improve this bound. We will investigate this in the following sections.

## 6 Faster Rates

Our framework can be applied with only the gradient-dominance property of each player's loss function. However, with different assumptions, the algorithm's convergence rate can be improved by utilizing different properties. The two properties we will investigate are smoothness and strong gradient-dominance.

## 6.1 Smoothness

As in Agarwal et al. (2019), for the direct parameterization, we have that the Value Function is smooth with respect to the $\pi$-player as in $V_W^\pi(\mu) - V_W^{\pi'}(\mu) = \mathcal{O}\left(\frac{2\gamma R_{\max}|\mathcal{A}|}{(1-\gamma)^2}\|\pi - \pi'\|_2\right)$.

---

**Algorithm 5:** Algorithm under Smoothness

**Data:** $T$

**for** $t \in [T]$ **do**

    Sample $\sigma_t \sim \text{Exp}(\eta)$;

    $\pi_t \leftarrow \mathcal{O}_\alpha\left(\sum_{i=1}^{t-1} -V_{W_i}^{\pi_t}(\mu) - \langle\sigma_t, \pi_t\rangle - V_{W_{t-1}}^{\pi_t}(\mu)\right)$ /* Optim.  Follow the Perturbed Leader    */

    $W_t \leftarrow \mathcal{O}_\alpha\left(\sum_{i=1}^{t} V_{W_t}^{\pi_i}(\mu) - \langle W_t, \sigma_t\rangle\right)$ /* Follow the Perturbed Leader Plus                     */

**end**

---

If our objective function is the value function and we can sufficiently demonstrate that the difference of value functions under different transition dynamics is smooth, we can improve our bounds by exploiting this smoothness. Formally, such smoothness would take the form of Condition 6.1.

**Condition 6.1.** *The difference in value functions between subsequent rounds is smooth with respect to policies such that for all $s \in \mathcal{S}$ and policies $\pi$ and $\pi'$, we have that*

$$[V_W^\pi(\mu) - V_{W'}^\pi(\mu)] - [V_W^{\pi'}(\mu) - V_{W'}^{\pi'}(\mu)] \le \tilde{L}\|W - W'\|_1\|\pi - \pi'\|_1.$$

In general, it is difficult to show that such an $\tilde{L}$ is smaller than the value that of the Lipschitz constant of the Value Function, $\frac{2\gamma R_{\max}|\mathcal{A}|}{(1-\gamma)^2}$. However, if we are in a setting such that $\tilde{L}$ is small, we can take advantage of this by using Optimistic Follow the Perturbed Leader Plus for the $\pi$-player where the optimistic function is simply the last loss function $h_{t-1}$. If Condition 6.1 holds, we have that $h_t - h_{t-1}$ is very smooth, and we can directly improve our rate of convergence. However, from the formulation of Condition 6.1, we also need to choose an online strategy for the environment such that $\|W - W'\|_1$ is bounded, which we can demonstrate is the case if the $W$-player uses FTPL+. We summarize these algorithmic choices in Algorithm 5. In the direct parameterization setting with the traditional objective function, we also show that the loss functions seen by OFTPL for the policy player and FTPL+ satisfy our gradient-dominance properties. In the direct parameterization case with traditional objective, these conditions hold as shown in the appendix respectively (see Lemma B.1 and Lemma B.2). Furthermore, in this setting, we have the robustness as such. Indeed, if $\tilde{L}$ is small here, we can improve the convergence and robustness of our trained algorithm.

**Theorem 6.1.** *Assume that the set of possible transition matrices $\mathcal{W}$ is convex and we are in the direct parameterization setting. Let $OL^\pi$ use OFTPL and $OL^W$ use FTPL+. Given that Condition 6.1 holds, we have that the robustness of the algorithm is for any $\bar{\pi}$ when setting $\eta = \sqrt{\frac{20L_\pi(d_w D_W + d_\pi D_\pi)^2}{d_\pi}D_\pi \tilde{L}_2 \alpha T}$,*

$$\min_{W^* \in \mathcal{W}} \sum_{t=0}^{T} V_{W^*}^{\bar{\pi}}(\mu) - \mathbb{E}\left[\min_{W^* \in \mathcal{W}} \sum_{t=0}^{T} V_{W^*}^{\pi_t}(\mu)\right] \le \mathcal{O}\left(\left[\frac{d_\pi \tilde{L}\sqrt{D_\pi(d_w D_w + d_\pi D_\pi)}}{\sqrt{5L_\pi T c_\pi(T_\mathcal{O}, K_\pi)}} + 1\right]c_\pi(T_\mathcal{O}, K_\pi) + c_W(T_\mathcal{O}, \mathcal{K}_W)\right).$$

## 6.2 Strong Gradient-Dominance

We now explore another extension: strong gradient-dominance. Moreover, in specific parameterizations, the objective functions for Robust MDPs will obey strong gradient-dominance, also known famously as the Polyak-Lojasiewicz condition (Polyak, 1963). This condition helps improve convergence in nonconvex settings, analogous to the strong convexity condition. We formally state such a property in Condition 6.2

**Condition 6.2.** *A function $a(x)$ satisfies $\mathcal{K}, \delta$-strong gradient-dominance if for any point $x \in \mathcal{X}$,*

$$\min_{x^* \in \mathcal{X}} a(x^*) \ge a(x) + \min_{x' \in \mathcal{X}}\left[\mathcal{K}\langle x' - x, \nabla a(x)\rangle + \frac{\delta}{2}\|x' - x\|_2^2\right].$$

In general, this is useful for achieving even tighter optimization bounds. Indeed, from Karimi et al. (2016), if we have this, Projected Gradient Descent enjoys better convergence.

**Lemma 6.1.** *Here, $x^* = \underset{x^* \in \mathcal{X}}{\arg\min}\, a_t(x^*)$ is the minimizer of $a_t$. $c_x^s$ is used for brevity. Given $a_t$ is $L$-Lipschitz continuous and is $\mathcal{K}, \delta$-strongly dominated, we have that using projected gradient descent gets global linear convergence as in*

$$a_t(x_k) - a_t(x^*) \le \left(1 - \frac{\delta}{\mathcal{K}^2 L}\right)^k (a_t(x_0) - a_t(x^*)) = c_x^s(k, \delta, \mathcal{K}).$$

## 6.3 Direct Parameterization with Regularization

Say we are in a setting where we want to maximize $V_W^\pi$ and ensure that the policy $\pi$ is regularized according to the $\ell_2$ norm. In this way, we can redefine the loss function to be $g(W, \pi) = V_W^\pi(\mu) - \|\pi\|_2^2$. Again, the sets $\mathcal{W}$ and $\mathcal{T}$ are bounded convex sets in this setting.

---

**Algorithm 6:** Algorithm under Regularization

**Data:** $T$

**for** $t \in [T]$ **do**

  Sample $\sigma_t \sim \text{Exp}(\eta)$;

  $\pi_t \leftarrow \mathcal{O}_\alpha \left( \sum_{i=1}^{t-1} -V_{W_i}^{\pi_t}(\mu) + \|\pi_t\|_2^2 - \langle \sigma_t, \pi_t \rangle \right)$ /* Follow the Perturbed Leader     */

  $W_t \leftarrow \mathcal{O}_\alpha \left( V_{W_t}^{\pi_t}(\mu) + \|\pi_t\|_2^2 \right)$ /* Best-Response     */

**end**

---

**Lemma 6.2.** *We have $g(W, \pi) = V_W^\pi(\mu) - \|\pi\|_2^2$ is $\mathcal{K}_\pi, \frac{T}{2}$ strongly gradient-dominated on set $\mathcal{T}$*

In this setting, if the $\pi$ player employs FTPL where its Approximate Optimization Oracle enjoys even better convergence and the $W$-Player employs Best Response, we have the robustness bound as follows. We summarize this algorithm in Algorithm 6. Indeed, given strong gradient-dominance, we have that the dependence on the complexity for the Optimization Oracle is better for the $\pi$-player, slightly improving the robustness bounds.

**Theorem 6.2.** *We are in the direct parameterization setting where the objective function is the regularized Value Function. Assume that the set of possible transition matrices $\mathcal{W}$ is convex. Let $OL^\pi$ use FTPL and $OL^W$ use Best-Response. Under direct parameterization, given that Condition 6.2 holds and $\eta = \frac{1}{L_\pi \sqrt{T d_\pi}}$, we have that the robustness of Algorithm 1 is for any $\bar{\pi}$*

$$\min_{W^* \in \mathcal{W}} \sum_{t=0}^T V_{W^*}^{\bar{\pi}}(\mu) - \|\bar{\pi}\|^2 - \min_{W^* \in \mathcal{W}} \sum_{t=0}^T V_{W^*}^{\pi_t}(\mu) 0 \|\pi_t\|^2 \le \frac{2 d_\pi^{\frac{3}{2}} D_\pi L_\pi}{\sqrt{T}} + c_\pi^s \left( T_\mathcal{O}, \mathcal{K}_\pi, \frac{1}{2} \right) + c_W(T_\mathcal{O}, \mathcal{K}_W).$$

# 7 Experiments

We now turn to explore how our algorithm finds robust minima empirically. We will use Algorithm 4 to optimize a policy in the GridWorld MDP (Sutton & Barto, 2018). This setting is a traditional MDP where the world is a grid where the initial state is one corner of the grid. The goal state is the opposite corner of the grid. At each step, the policy can take any of four actions. The next state is sampled respectively from the transition matrix. If the policy lands in the goal state, it receives a reward of 10, and the MDP is finished. Otherwise, it receives a reward of $-1$. We experiment in the setting where the grid is 5 states tall and 5 states wide. We wish to measure how quickly the robustness of policy is improved through each iteration of our algorithm. As a metric to measure robustness, given a policy, we choose the transition matrix that minimizes the expected reward of the initial state and reports the initial state's expected reward. We do this for every iteration of our algorithm. We will do this over different adversarial transition matrix sets. The sets in question will be

$$\mathcal{T} = \{T \text{ s.t. } \|T - T_0\|_q \le \tau\}.$$

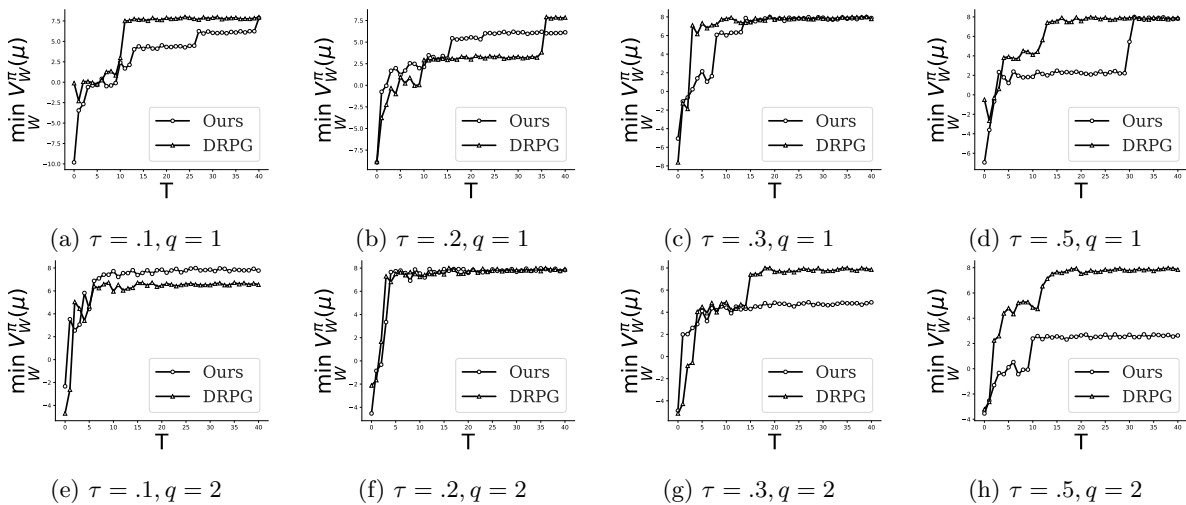

Figure 1: We plot the convergence of our algorithm over many different transition matrix uncertainty set shapes. We see that over all shapes, our algorithm converges in roughly the predicted $\frac{1}{\sqrt{T}}$ rate predicted by our results. The convergence curves of DRPG and our algorithms are very similar.

Here, $T_0$ is some randomly generated initial transition matrix, $q$ is a hyperparameter affecting the shape of the transition set, and $\tau$ is the radius of the transition set. We demonstrate robustness improvement over several values of $\tau$ and $q$. We plot the convergence of our algorithm over $q \in \{1, 2\}$ and $\tau \in \{.1, .2, .3, .5\}$ in Figure 1. As a baseline, we provide the convergence results for Double-Loop Robust Policy Gradient (DRPG) from Wang et al. (2022b). We see in these convergence plots that our algorithm roughly achieves a convergence rate of $\mathcal{O}\left(T^{-\frac{1}{2}}\right)$. This is achieved across all uncertainty set shapes and radii. This corroborates the theoretical results of our framework's versatility and efficiency in different environmental settings. When comparing our results to the baseline, we see that our method and DRPG achieve similar convergence curves. Overall, these results suggest that, empirically, this algorithm performs competitively with existing algorithms and slightly.

## 7.1 Implementation of the Practical Algorithm

We have provided a GitHub repository to reproduce our experiments. We use the OpenAI Gym environment to set up our experiments. For all optimization tasks, we utilitize the constrained optimization libraries in SciPy. Moreover, implementing projected gradient descent unfortunately requires manually designing the gradient calculation depending on the problem setting.

## 8 Discussion and Limitations

In this paper, we developed a nonconvex No-Regret framework that has decoupled the convergence for Robust MDP algorithms. Based on the proposed framework, we designed different Robust MDP algorithms under standard gradient-dominance, strong gradient-dominance, and smooth MDPs. The proven convergence results are some of the strongest in the literature, with only a convexity assumption on the set of possible transition matrices. Possible extensions include using this nonconvex No-Regret Framework for other nonconvex problems, such as other nonconvex games, or exploring how different minimization oracles could empirically improve the performance of our algorithms. Another possible avenue could be studying Robust MDPs where the optimization objective obeys the strict saddle property.

**Limitations** However, our work has several limitations. Firstly, we require some gradient-dominance conditions for the policy and environmental dynamics player. In many settings, the objective function for the Robust MDP does not satisfy this. This assumption does reduce the scope of our work. Moreover, many of our convergence guarantees are only made in expectation, while many other bounds in the literature are

made absolutely. Moreover, we do not generate a single policy that achieves strong robustness but, instead, a series of policies that, if used together, obey our convergence guarantees.

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

| Symbol | Meaning |
|---|---|
| $\gamma$ | Discount Factor |
| $\mathcal{S}$ | Set of Actions |
| $\mathbb{P}_W$ | Transition Matrix parameterized by $W$ |
| $\mathcal{R}$ | Reward Function |
| $W$ | Parameter for Environment |
| $\mathcal{R}_{\max}$ | Maximum Reward |
| $\theta$ | Parameter for the policy |
| $\mathcal{T}$ | Set of Policy Parameters |
| $\pi_\theta, \pi$ | Policy |
| $\mu$ | Distribution over starting states |
| $V_W^\pi$ | Value Function |
| $d_{s_0}^W, d_{s_0}^\pi$ | Occupancy Measure of state |
| $g$ | Objective function of Robust MDP |
| $l_t$ | Loss Function for Environmental Player |
| $h_t$ | Loss Function for Policy Player |
| $\mathrm{Reg}_W$ | Regret Of Environment Player |
| $\mathrm{Reg}_\pi$ | Regret Of Policy Player |
| $\mathcal{O}_\alpha$ | Optimization Oracle |
| $\sigma$ | Noise Vector for Online Algorithms |
| $\eta$ | Parameter for the Noise Distribution |
| $\mathcal{K}_W, \mathcal{K}_\pi$ | Gradient Dominance Constant for $W$ and $\pi$ |
| $T_{\mathcal{O}}$ | Number of Iterations for Optimization Oracle |
| $c_x(T_{\mathcal{O}}, \mathcal{K})$ | Suboptimality of Projected Gradient Descent |
| $L_\pi$ | Lipschitz Smoothness Constant of $h_t$ |
| $D_\pi$ | Radius of Set of Policies |
| $d_\pi$ | Dimension of $\theta$ |
| $c_W(T_{\mathcal{O}}, \mathcal{K}_W)$ | Suboptimality of Projected Gradient Descent for Environment Player |
| $c_\pi(T_{\mathcal{O}}, \mathcal{K}_\pi)$ | Suboptimality of Projected Gradient Descent for Policy Player |
| $\tilde{L}$ | Lipschitz Constant of Difference of Objective Functions |
| $\delta$ | Constant for Strong Gradient Dominance |
| $c_x^s(T_{\mathcal{O}}, \mathcal{K})$ | Suboptimality of Projected Gradient Descent under Strong Gradient Dominance |

Table 1: Table of Notation

# A    Proof of Preliminary Theorems

## A.1    Proof of Theorem 4.1

**Theorem 4.1.** *We have the difference between the robustnesses of the chosen policies and any policy $\bar{\pi}$ is upper bounded by the regret of the two players*

$$\frac{1}{T}\min_{W^* \in \mathcal{W}}\sum_{t=0}^{T} g(W^*, \bar{\pi}) - \frac{1}{T}\min_{W^* \in \mathcal{W}}\sum_{t=0}^{T} g(W^*, \pi_t) \leq Reg_W + Reg_\pi.$$

*Here, $Reg_W$ and $Reg_\pi$ are the two average regrets of the two players $OL^W$ and $OL^\pi$.*

*Proof.* By definition, the regret of the $W$-player is equivalent to

$$\text{Reg}_W = \sum_{t=0}^{T} l_t(W_t) - \min_{W^*} \sum_{t=0}^{T} l_t(W^*)$$

$$= \sum_{t=0}^{T} g(W_t, \pi_t) - \min_{W^*} \sum_{t=0}^{T} g(W^*, \pi_t)$$

Similarly, for the $\pi$ player, we have that

$$\text{Reg}_\pi = \sum_{t=0}^{T} l_t(\pi_t) - \min_{\pi^*} \sum_{t=0}^{T} l_t(\pi^*)$$

$$= \max_{\pi^*} \sum_{t=0}^{T} g(W_t, \pi^*) - \sum_{t=0}^{T} g(W_t, \pi_t) \tag{1}$$

Therefore, we can upper bound the sum of objective functions throughout our training process as

$$\sum_{t=0}^{T} g(W_t, \pi_t) = \text{Reg}_W + \min_{W^*} \sum_{t=0}^{T} g(W^*, \pi_t).$$

We similarly lower bound the sum of value functions throughout our training process as

$$\sum_{t=0}^{T} g(W_t, \pi_t) = \max_{\pi^*} \sum_{t=0}^{T} g(W_t, \pi^*) - \text{Reg}_\pi$$

$$\geq \sum_{t=0}^{T} g(W_t, \bar{\pi}) - \text{Reg}_\pi$$

$$\geq \min_{W^*} \sum_{t=0}^{T} g(W^*, \bar{\pi}) - \text{Reg}_\pi \tag{2}$$

Combining Equation (1) and Equation (2), we have our desired statement

$$\min_{W^*} \sum_{t=0}^{T} g(W^*, \bar{\pi}) - \min_{W^*} \sum_{t=0}^{T} g(W^*, \pi_t) \leq \text{Reg}_W + \text{Reg}_\pi.$$

$\square$

# B  Proofs of Gradient Dominance

## B.1  Proof of Lemma 5.2

**Lemma 5.2.** *The gradient of the value function $V^W$ with respect to the parameter $W$, denoted as $\nabla_W V^W(s)$, is given by*

$$\nabla_W V^W(s) = \frac{1}{1-\gamma} \sum_{s',a,s} d_\mu^W(s) \pi(a,s) \nabla \mathbb{P}_W(s',a,s) V^W(s').$$

*Proof.* We wish to calculate the gradient of the value function $V^W$ with respect to $\mathbb{P}_W(s', a, s)$. We have that

$$\nabla_W V^W(s) = \nabla_W \left( \sum_a \pi(a, s) Q_\pi(s, a) \right)$$

$$= \sum_a \pi(a, s) \nabla_W Q_\pi(s, a)$$

$$= \sum_a \pi(a, s) \nabla_W \left[ R(s, a) + \gamma \sum_{s'} \mathbb{P}_W(s', a, s) V^\pi(s') \right] \tag{3}$$

$$= \sum_a \gamma \pi(a, s) \sum_{s'} \left[ \nabla_W \mathbb{P}_W(s', a, s) V^\pi(s') + \mathbb{P}_W(s', a, s) \nabla_W V^\pi(s') \right]$$

$$= \sum_{a,s'} \gamma \pi(a, s) \mathbb{P}_W(s', a, s) \nabla_W V^\pi(s') + \sum_{s',a} \pi(a, s) \gamma \nabla_W \mathbb{P}_W(s', a, s) V^\pi(s')$$

Here, the Equation (3) comes from the definition of the $Q$ function. Unrolling this makes it such that we have

$$\nabla_W V^W(\mu) = \sum_{t=0}^{\infty} \sum_{s',a,s} Pr(s_t = s | \mu) \gamma^t \pi(a, s) \nabla \mathbb{P}_W(s', a, s) V^W(s')$$

$$= \frac{1}{1 - \gamma} \sum_{s',a,s} d_\mu^W(s) \pi(a, s) \nabla \mathbb{P}_W(s', a, s) V^W(s')$$

We have now arrived at our desired quantity. $\qquad\square$

## B.2 Proof of Lemma 5.3

**Lemma 5.3.** *Given two different transition dynamics parameters $W$ and $W'$, we have that*

$$V_W(\mu) - V_{W'}(\mu) = \sum_{s',a,s} d_\mu^W(s) \mathbb{P}_W(s', a, s) \pi(a, s) A^{W'}(s', a, s).$$

*Here, we define the $W$-Advantage Function as $A^W(s', a, s) = \gamma V_W(s') + r(s, a) - V_W(s)$.*

*Proof.* This proof follows mainly from the proof of the Performance Difference Lemma from Kakade & Langford (2002).

$$V_W(\mu) - V_{W'}(\mu) = \mathbb{E}_{\mathbb{P}_W, \pi} \sum_{t=0}^{\infty} \gamma^t r(s_t, a_t) - V^{W'}(\mu)$$

$$= \mathbb{E}_{\mathbb{P}_W, \pi} \left[ \sum_{t=0}^{\infty} \gamma^t r(s_t, a_t) + \gamma^t V_{W'}(s_t) - \gamma^t V_{W'}(s_t) \right] - V^{W'}(\mu)$$

$$= \mathbb{E}_{\mathbb{P}_W, \pi} \left[ \sum_{t=0}^{\infty} \gamma^t r(s_t, a_t) + \gamma^{t+1} V_{W'}(s_{t+1}) - \gamma^t V_{W'}(s_t) \right]$$

$$= \mathbb{E}_{\mathbb{P}_W, \pi} \left[ \sum_{t=0}^{\infty} \gamma^t A^{W'}(s_{t+1}, a_t, s_t) \right]$$

$$= \frac{1}{1 - \gamma} \sum_{s',a,s} \left[ \gamma^t d_\mu^W(s) \mathbb{P}_W(s', a, s) \pi(a, s) A^{W'}(s', a_t, s) \right] \tag{4}$$

Here, Equation (4) comes from our definition of the Advantage function for the $W$-player. This concludes the proof. $\qquad\square$

## B.3 Proof of Lemma 5.5

**Lemma 5.5.** *Under direct parameterization, we have that the $V_W(\mu)$ is gradient-dominated with constant $\mathcal{K}_W = \frac{1}{1-\gamma} \left\| \frac{d_\mu^{W^*}}{\mu} \right\|_\infty$ as in for an arbitrary $W \in \mathcal{W}$ and the optimal parameter $W^* \in \mathcal{W}$, we have*

$$V_W(\mu) - V_{W^*}(\mu) \leq -\mathcal{K}_W \min_{\bar{W} \in \mathcal{W}} \left[ \left( \bar{W} - W \right)^\top \nabla_W V_W(\mu) \right].$$

*Here, we have that Subcondition 4 holds for the $W$-player.*

*Proof.* We can use Lemma 5.3 to prove this. We will lower bound the difference between the optimal $V_{W^*}(\mu)$ and the $V_W(\mu)$. In order to prove Gradient Dominance, we need that $V_W(\mu) - V_{W^*}(\mu)$ to be upper-bounded. We will equivalently lower bound the negative of this. We have

$$V_W(\mu) - V_{W^*}(\mu) = \frac{-1}{1-\gamma} \sum_{s',a,s} \left[ \gamma^t d_\mu^{W^*}(s) \pi(a,s) \mathbb{P}_{W^*}(s'|a,s) A^W(s',a,s) \right]$$

$$\leq \frac{-1}{1-\gamma} \sum_{s',a,s} \left[ \gamma^t d_\mu^{W^*}(s) \pi(a,s) \min_{s'} \left( A^W(s',a,s) \right) \right] \tag{5}$$

$$\leq \left( \max_s \frac{d_\mu^{W^*}(s)}{d_\mu^W(s)} \right) \frac{-1}{1-\gamma} \sum_{s',a,s} \left[ \gamma^t d_\mu^W(s) \pi(a,s) \min_{s'} \left( A^W(s',a,s) \right) \right]$$

Here, the Equation (5) comes from seeing that the value $\mathbb{P}_{W^*}(s'|a,s) A^W(s',a,s)$ is minimized when $\mathbb{P}_{W^*}$ puts the most weight on the state minimizing the advantage function. Looking only at that last term, we can bound it in the following manner

$$\frac{-1}{1-\gamma} \sum_{s',a,s} \left[ \gamma^t d_\mu^W(s) \pi(a,s) \min_{s'} \left( A^W(s',a,s) \right) \right]$$

$$= \frac{-1}{1-\gamma} \min_{\bar{W}} \sum_{s',a,s} \left[ \gamma^t d_\mu^W(s) \pi(a,s) \, \mathbb{P}_{\bar{W}}(s',a,s) \left( A^W(s',a,s) \right) \right] \tag{6}$$

$$= \frac{-1}{1-\gamma} \min_{\bar{W}} \sum_{s',a,s} \left[ \gamma^t d_\mu^W(s) \pi(a,s) \, \left( \mathbb{P}_{\bar{W}}(s',a,s) - \mathbb{P}_W(s',a,s) \right) \left( A^W(s',a,s) \right) \right] \tag{7}$$

$$= \frac{-1}{1-\gamma} \min_{\bar{W}} \sum_{s',a,s} \left[ \gamma^t d_\mu^W(s) \pi(a,s) \, \left( \mathbb{P}_{\bar{W}}(s',a,s) - \mathbb{P}_W(s',a,s) \right) \left( V_W(s') \right) \right] \tag{8}$$

$$= -\min_{\bar{W}} \left[ \left( \bar{W} - W \right)^\top \nabla_{W'} V_W(\mu) \right] \tag{9}$$

Here, Equation (6) comes from seeing that the value $\mathbb{P}_{W^*}(s'|a,s) A^W(s',a,s)$ is minimized when $\mathbb{P}_{W^*}$ puts the most weight on the state minimizing the advantage function. Equation (7) comes from the fact that the $\sum_{s'} \mathbb{P}_W(s',a,s) A^W(s',a,s) = 0$. Equation (8) comes from the definition of the $W$-player advantage function. Finally, Equation (9) comes from Lemma 5.2. Combining these yield

$$V_W(\mu) - V_{W^*}(\mu) \leq - \left\| \frac{d_\mu^{W^*}}{d_\mu^W} \right\|_\infty \min_{\bar{W}} \left[ \left( \bar{W} - W \right)^\top \nabla_{W'} V_W(\mu) \right]$$

$$\leq \frac{-1}{1-\gamma} \left\| \frac{d_\mu^W}{\mu} \right\|_\infty \min_{\bar{W}} \left[ \left( \bar{W} - W \right)^\top \nabla_{W'} V_W(\mu) \right] \tag{10}$$

Equation (10) comes from the fact that $d_\mu^{W^*}(s) \geq (1-\gamma)\mu(s)$ by definition. Here, flipping this, we have

$$V_W(\mu) - V_{W^*}(\mu) \leq \frac{-1}{1-\gamma} \left\| \frac{d_\mu^{W^*}}{\mu} \right\|_\infty \min_{\bar{W}} \left[ \left( \bar{W} - W \right)^\top \nabla_{W'} V_W(\mu) \right]$$

This is a satisfying definition of gradient dominance. □

## B.4 Proof of Lemma 5.4

**Lemma 5.4.** *Let the objective function be the Value Function. For any positive noise term $\sigma$, we have that under direct parameterization, $\sum_i^{t-1} l_i(\cdot) - \langle \sigma, \cdot \rangle$ and $\sum_i^{t-1} h_i(\cdot) - \langle \sigma, \cdot \rangle$ are both gradient-dominated for the $W$-player and the $\pi$-player on sets $\mathcal{W}$ and $\mathcal{T}$ with constants $\mathcal{K}_W = \frac{1}{1-\gamma}\left\|\frac{d_\mu^{W^*}}{\mu}\right\|_\infty$ and $\mathcal{K}_\pi = \frac{1}{1-\gamma}\left\|\frac{d_\mu^{\pi^*}}{\mu}\right\|_\infty$ respectively. Therefore, Subcondition 2 hold for both the loss functions for both players.*

*Proof.* We can use Lemma 5.3 to prove both. We start with the $W$ player.

$$\sum_i^{t-1} V_W^{\pi_i}(\mu) - V_{W^*}^{\pi_i}(\mu) - \langle \sigma, W - W^* \rangle =$$

$$\sum_i^{t-1} \frac{-1}{1-\gamma} \sum_{s',a,s} \left[ d_\mu^{W^*}(s)\pi_i(a|s)\mathbb{P}_{W^*}(s'|a,s)A^W(s',a,s) \right] - \langle \sigma, W - W^* \rangle \tag{11}$$

$$\leq \left( \max_s \frac{d_\mu^{W^*}(s)}{d_\mu^W(s)} \right) \left[ \sum_i^{t-1} \frac{-1}{1-\gamma} \sum_{s',a,s} \left[ d_\mu^W(s)\pi_i(a|s)\, \mathbb{P}_{W^*}(s'|a,s)\left(A^W(s',a,s)\right) \right] \right.$$

$$\left. - \langle \sigma, W - W^* \rangle \right]$$

$$\leq \left( \max_s \frac{d_\mu^{W^*}(s)}{d_\mu^W(s)} \right) - \min_{\bar{W}} \left[ \sum_i^{t-1} \frac{1}{1-\gamma} \sum_{s',a,s} \left[ d_\mu^{W^*}(s)\pi_i(a|s)\mathbb{P}_{\bar{W}}(s'|a,s)\left(A^W(s',a,s)\right) \right] \right. \tag{12}$$

$$\left. - \langle \sigma, \bar{W} - W \rangle \right]$$

Here, Equation (11) comes from the fact that the maximum of the interior of the RHS is always nonnegative, and Equation (12) comes from using the minimizing transition dynamics $\bar{W}$. Looking at the inside term, we have that

$$-\min_{\bar{W}} \left[ \frac{-1}{1-\gamma} \sum_{s',a,s} \sum_i \left[ d_\mu^W(s)\pi_i(a|s)\, \mathbb{P}_{\bar{W}}(s',a,s)\left(A^W(s',a,s)\right) \right] - \langle \sigma, \bar{W} - W \rangle \right]$$

$$= \min_{\bar{W}} \left[ \frac{-1}{1-\gamma} \sum_{s',a,s} \sum_i \left[ d_\mu^W(s)\pi_i(a|s)\, \left(\mathbb{P}_{\bar{W}}(s',a,s) - \mathbb{P}_W(s',a,s)\right)\left(A^W(s',a,s)\right) \right] \right. \tag{13}$$

$$\left. - \langle \sigma, \bar{W} - W \rangle \right]$$

$$= -\min_{\bar{W}} \left[ \frac{-1}{1-\gamma} \sum_{s',a,s} \sum_i \left[ d_\mu^W(s)\pi_i(a|s)\, \left(\mathbb{P}_{\bar{W}}(s',a,s) - \mathbb{P}_W(s',a,s)\right)\left(V_W(s')\right) \right] \right. \tag{14}$$

$$\left. - \langle \sigma, \bar{W} - W \rangle \right]$$

$$= -\min_{\bar{W}} \left[ \left(\bar{W} - W\right)^\top \nabla_W \left[ \sum_i V_W^{\pi_i}(\mu) - \langle \sigma, W \rangle \right] \right] \tag{15}$$

Equation (13) comes from the fact that the $\sum_{s'} \mathbb{P}_W(s',a,s)A^W(s',a,s) = 0$. Equation (14) comes from the definition of the $W$-player advantage function. Finally, comes from Lemma 5.2. Combining these, we have

that

$$\sum_i^{t-1} V_W^{\pi_i}(\mu) - V_{W^*}^{\pi_i}(\mu) - \langle \sigma, W - W^* \rangle \leq$$

$$\left\| \frac{d_\mu^{W^*}}{\mu} \right\|_\infty \frac{-1}{1-\gamma} \min_{\bar{W}} \left[ (W - \bar{W})^\top \nabla_W \left[ \sum_i V_W^{\pi_i}(\mu) - \langle \sigma, W \rangle \right] \right]$$

Moreover, we use the fact that $d_\mu^W(s) \geq (1-\gamma)\mu(s)$ by definition. We now do this for the $\pi$-player. By the Performance Difference Lemma,

$$\sum_i^{t-1} V_{W_i}^{\pi^*} - V_{W_i}^{\pi} - \langle \sigma, \pi - \pi^* \rangle = \frac{1}{1-\gamma} \sum_i^{t-1} \sum_{s,a} d_\mu^{\pi^*}(s)\pi^*(a,s)A^\pi(s,a) - \langle \sigma, \pi - \pi^* \rangle \tag{16}$$

$$\leq \frac{-1}{1-\gamma} \min_{\bar{\pi}} \left[ -\sum_i^{t-1} \sum_{s,a} d_\mu^{\pi^*}(s)\bar{\pi}(s,a)A^\pi(s,a) - \langle \sigma, \bar{\pi} - \pi \rangle \right]$$

$$\leq - \left\| \frac{d_\mu^{\pi^*}}{d_\mu^\pi} \right\|_\infty \min_{\bar{\pi}} \left[ \frac{-1}{1-\gamma} \sum_i^{t-1} \sum_{s,a} d_\mu^\pi(s)\bar{\pi}(s,a)A^\pi(s,a) - \langle \sigma, \bar{\pi} - \pi \rangle \right]$$

$$= - \left\| \frac{d_\mu^{\pi^*}}{d_\mu^\pi} \right\|_\infty \min_{\bar{\pi}} \left[ \frac{-1}{1-\gamma} \sum_i^{t-1} \sum_{s,a} d_\mu^\pi(s)(\bar{\pi}(s,a) - \pi(a,s))A^\pi(s,a) - \langle \sigma, \bar{\pi} - \pi \rangle \right] \tag{17}$$

$$= - \left\| \frac{d_\mu^{\pi^*}}{d_\mu^\pi} \right\|_\infty \min_{\bar{\pi}} \left[ \frac{-1}{1-\gamma} \sum_i^{t-1} \sum_{s,a} d_\mu^\pi(s)(\bar{\pi}(s,a) - \pi(a,s))Q^\pi(s,a) - \langle \sigma, \bar{\pi} - \pi \rangle \right] \tag{18}$$

$$\leq \frac{-1}{1-\gamma} \left\| \frac{d_\mu^{\pi^*}}{\mu} \right\|_\infty \min_{\bar{\pi}} (\bar{\pi} - \pi)^\top \nabla_\pi \left( -\sum_i^{t-1} V_{W_i}^\pi - \langle \sigma, \pi \rangle \right) \tag{19}$$

Here, Equation (16) comes from the Performance Difference Lemma, Equation (17) comes from the fact that $\sum_a \pi(a,s)A^\pi(s,a) = 0$, Equation (18) comes from the definition of the Advantage Function for the $\pi$-player, and Equation (19) comes from the both the Policy Gradient Theorem and the fact that $d_\mu^\pi(s) \geq (1-\gamma)\mu(s)$. We now have proven both claims of our lemma. □

**Lemma B.1.** *The $\pi$-player enjoys Subcondition 3, i.e. $\sum_i^{t-1} h_i(\cdot) + h_{t-1}(\cdot) - \sigma$ is gradient-dominated with constant $\frac{1}{1-\gamma} \left\| \frac{d_\mu^{\pi^*}}{\mu} \right\|_\infty$.*

*Proof.* For simplicity, we will call $\sum_j h_j(\cdot) := \sum_i^{t-1} h_i(\cdot) + h_{t-1}(\cdot)$ where $j$ indexes over the set $\{h_1, \ldots, h_{t-2}, h_{t-1}, h_{t-1}\}$. With this, we can follow through with our proof. By the performance difference

lemma,

$$\sum_j V_{W_i}^{\pi^*} - V_{W_i}^{\pi} - \langle \sigma, \pi - \pi^* \rangle$$

$$= \frac{1}{1-\gamma} \sum_j \sum_{s,a} d_\mu^{\pi^*}(s) \pi^*(a,s) A^\pi(s,a) - \langle \sigma, \pi^* - \pi \rangle \tag{20}$$

$$\leq \frac{-1}{1-\gamma} \min_{\bar{\pi}} \left[ -\sum_j \sum_{s,a} d_\mu^{\pi^*}(s) \bar{\pi}(s,a) A^\pi(s,a) - \langle \sigma, \bar{\pi} - \pi \rangle \right]$$

$$\leq - \left\| \frac{d_\mu^{\pi^*}}{d_\mu^\pi} \right\|_\infty \min_{\bar{\pi}} \left[ \frac{-1}{1-\gamma} \sum_j \sum_{s,a} d_\mu^{\pi}(s) \bar{\pi}(s,a) A^\pi(s,a) - \langle \sigma, \bar{\pi} - \pi \rangle \right]$$

$$= - \left\| \frac{d_\mu^{\pi^*}}{d_\mu^\pi} \right\|_\infty \min_{\bar{\pi}} \left[ \frac{-1}{1-\gamma} \sum_j \sum_{s,a} d_\mu^{\pi}(s) (\bar{\pi}(s,a) - \pi(a,s)) A^\pi(s,a) - \langle \sigma, \bar{\pi} - \pi \rangle \right] \tag{21}$$

$$= - \left\| \frac{d_\mu^{\pi^*}}{d_\mu^\pi} \right\|_\infty \min_{\bar{\pi}} \left[ \frac{-1}{1-\gamma} \sum_j \sum_{s,a} d_\mu^{\pi}(s) (\bar{\pi}(s,a) - \pi(a,s)) Q^\pi(s,a) - \langle \sigma, \bar{\pi} - \pi \rangle \right] \tag{22}$$

$$\leq \frac{1}{1-\gamma} \left\| \frac{d_\mu^{\pi^*}}{\mu} \right\|_\infty \min_{\bar{\pi}} (\bar{\pi} - \pi)^\top \nabla_\pi \left( \sum_j -V_{W_i}^{\pi} - \langle \sigma, \pi \rangle \right) \tag{23}$$

Here, Equation (20) comes from the Performance Difference Lemma, Equation (21) comes from the fact that $\sum_a \pi(a,s) A^\pi(s,a) = 0$, Equation (22) comes from the definition of the Advantage Function for the $\pi$-player, and Equation (23) comes from the both the Policy Gradient Theorem and the fact that $d_\mu^\pi(s) \geq (1-\gamma)\mu(s)$. $\quad\square$

**Lemma B.2.** *Subcondition 1 is satisfied for the $W$-player, i.e. $\sum_i^t l_i - \sigma$ is gradient dominated.*

*Proof.* We can use Lemma 5.3 to prove both. We start with the $W$ player.

$$\sum_i^t V_W^{\pi_i}(\mu) - V_{W^*}^{\pi_i}(\mu) - \langle \sigma, W - W^* \rangle =$$

$$\sum_i^t \frac{-1}{1-\gamma} \sum_{s',a,s} \left[ d_\mu^{W^*}(s) \pi_i(a|s) \mathbb{P}_{W^*}(s'|a,s) A^W(s',a,s) \right] - \langle \sigma, W - W^* \rangle$$

$$\leq \left\| \frac{d_\mu^{W^*}(s)}{d_\mu^W(s)} \right\|_\infty \left[ \sum_i^t \frac{-1}{1-\gamma} \sum_{s',a,s} \left[ d_\mu^W(s) \pi_i(a|s) \, \mathbb{P}_{W^*}(s'|a,s) \left( A^W(s',a,s) \right) \right] \right.$$

$$\left. - \langle \sigma, W - W^* \rangle \right]$$

$$\leq - \left\| \frac{d_\mu^{W^*}(s)}{d_\mu^W(s)} \right\|_\infty \left[ \min_{\bar{W}} \sum_i^t \frac{1}{1-\gamma} \sum_{s',a,s} \left[ d_\mu^{W^*}(s) \pi_i(a|s) \mathbb{P}_{\bar{W}}(s'|a,s) \left( A^W(s',a,s) \right) \right] \right. \tag{24}$$

$$\left. - \langle \sigma, \bar{W} - W \rangle \right]$$

Here, Equation (24) comes from using the minimizing transition dynamics $\bar{W}$. Looking at the inside term, we have that

$$\min_{\bar{W}} \left[ \frac{1}{1-\gamma} \sum_{s',a,s} \sum_i \left[ d_\mu^W(s) \pi_i(a|s) \, \mathbb{P}_{\bar{W}}(s',a,s) \left( A^W(s',a,s) \right) \right] - \langle \sigma, \bar{W} - W \rangle \right]$$

$$= \min_{\bar{W}} \left[ \frac{1}{1-\gamma} \sum_{s',a,s} \sum_i \left[ d_\mu^W(s) \pi_i(a|s) \, \left( \mathbb{P}_{\bar{W}}(s',a,s) - \mathbb{P}_W(s',a,s) \right) \left( A^W(s',a,s) \right) \right] \right. \tag{25}$$

$$\left. - \langle \sigma, \bar{W} - W \rangle \right]$$

$$= \min_{\bar{W}} \left[ \frac{1}{1-\gamma} \sum_{s',a,s} \sum_i \left[ d_\mu^W(s) \pi_i(a|s) \, \left( \mathbb{P}_{\bar{W}}(s',a,s) - \mathbb{P}_W(s',a,s) \right) \left( V_W(s') \right) \right] \right. \tag{26}$$

$$\left. - \langle \sigma, \bar{W} - W \rangle \right]$$

$$= \min_{\bar{W}} \left[ \left( \bar{W} - W \right)^\top \nabla_W \left[ \sum_i V_W^{\pi_i}(\mu) - \langle \sigma, W \rangle \right] \right] \tag{27}$$

Equation (25) comes from the fact that the $\sum_{s'} \mathbb{P}_W(s',a,s) A^W(s',a,s) = 0$. Equation (26) comes from the definition of the $W$-player advantage function. Finally, Equation (27) comes from Lemma 5.2. Combining these, we have that

$$\sum_i V_W^{\pi_i}(\mu) - V_{W^*}^{\pi_i}(\mu) - \langle \sigma, W - W^* \rangle \leq$$

$$\frac{-1}{1-\gamma} \left\| \frac{d_\mu^{W^*}}{\mu} \right\|_\infty \min_{\bar{W}} \left[ \left( \bar{W} - W \right)^\top \nabla_W \left[ \sum_i V_W^{\pi_i}(\mu) - \langle \sigma, W \rangle \right] \right]$$

Moreover, we use the fact that $d_\mu^W(s) \geq (1-\gamma)\mu(s)$ by definition. $\qquad\square$

## C  Proofs for Convergence

Here, we detail a lemma on the convergence of FTPL and OFTPL as proven in Suggala & Netrapalli (2019).

**Lemma C.1.** *Let $D$ be the $\ell_\infty$ diameter of the space $\mathcal{X}$. Suppose the losses encountered by the learner are $L$-Lipschitz w.r.t $\ell_1$ norm. Moreover, suppose the optimization oracle used has error $\alpha$. For any fixed $\eta$, the predictions of Follow the Perturbed Leader satisfy the following regret bound. Here, $d$ is the dimension of the noise vector.*

$$\mathbb{E} \left[ \frac{1}{T} \sum_{t=1}^T f_t(x_t) - \frac{1}{T} \inf_{x \in \mathcal{X}} \sum_{t=1}^T f_t(x) \right] \leq O \left( \eta d^2 D L^2 + \frac{dD}{\eta T} + \alpha \right).$$

*For OFTPL, suppose our guess $g_t$ is such that $g_t - f_t$ is $L_t$-Lipschitz w.r.t $\ell_1$ norm, for all $t \in [T]$. The predictions of Optimistic Follow the Perturbed Leader satisfy the following regret bound.*

$$\mathbb{E} \left[ \frac{1}{T} \sum_{t=1}^T f_t(x_t) - \frac{1}{T} \inf_{x \in \mathcal{X}} \sum_{t=1}^T f_t(x) \right] \leq O \left( \eta d^2 D \sum_{t=1}^T \frac{L_t^2}{T} + \frac{dD}{\eta T} + \alpha \right).$$

### C.1  Proof of Lemma 4.1

**Lemma 4.1.** *Suppose we have an incoming sequence of loss functions $f_t$ for $t \in [T]$ with an optimization oracle that can minimize a function to less than $\alpha$ error. The Best Response algorithm satisfies the following regret bound*

$$\frac{1}{T} \sum_{t=1}^T f_t(x_t) - \frac{1}{T} \inf_{x \in \mathcal{X}} \sum_{t=1}^T f_t(x) \leq \alpha.$$

*Proof.* The regret term is defined as $\frac{1}{T} \sum_{t=1}^{T} f_t(x_t) - \frac{1}{T} \inf_{x \in \mathcal{X}} \sum_{t=1}^{T} f_t(x)$ where $x_t$ are the choices taken by the BestResponse algorithm. For an arbitrary time step $t$, we have that

$$f_t(x_t) - f_t(x) \leq \min_{x^*} f_t(x^*) + \alpha - f_t(x) \tag{28}$$

$$\leq \alpha$$

where Equation (28) comes from the fact that an optimization oracle is used to calculate $x_t$ and has error upper bounded by $\alpha$. Therefore, we have that

$$\frac{1}{T} \sum_{t=1}^{T} f_t(x_t) - \frac{1}{T} \inf_{x \in \mathcal{X}} \sum_{t=1}^{T} f_t(x) \leq \alpha.$$

$\square$

# D   Proofs for Extension Section

## D.1   Proof of Lemma D.1

**Lemma D.1.** *Given a series of choices by FTPL+ $x_1, \ldots, x_T$ with smooth and gradient dominated loss functions $f_1, \ldots, f_t$ and noise sampled $\sigma \sim Exp(\eta)$, the stability in choices is bounded in expectation by*

$$\mathbb{E}(\|x_{t+1} - x_t\|_1) \leq 125\eta L d^2 D + \frac{\alpha}{20L}$$

To prove this, we will first prove two properties of monotonicity of the loss function on an input of noise $\sigma$. Much of this proof structure is inspired by Suggala & Netrapalli (2019).

**Lemma D.2.** *Let $x_t(\sigma)$ be the solution chosen by FTPL+ under noise sigma. Let $\sigma' = \sigma + ce_i$ for some positive constant $c$, then we have that*

$$x_{t,i}(\sigma') \geq x_{t,i}(\sigma) - \frac{2\alpha}{c}.$$

*Proof.* Given that the approximate optimality of $x_t(\sigma)$, we have that

$$\sum_{i=1}^{t-1} f_i(x_t(\sigma)) + m_t(x_t(\sigma)) - \langle \sigma, x_t(\sigma) \rangle \tag{29}$$

$$\leq \sum_{i=1}^{t-1} f_i(x_t(\sigma')) + m_t(x_t(\sigma')) - \langle \sigma, x_t(\sigma') \rangle + \alpha \tag{30}$$

$$= \sum_{i=1}^{t-1} f_i(x_t(\sigma')) + m_t(x_t(\sigma')) - \langle \sigma', x_t(\sigma') \rangle + \langle \sigma' - \sigma, x_t(\sigma') \rangle + \alpha$$

$$\leq \sum_{i=1}^{t-1} f_i(x_t(\sigma)) + m_t(x_t(\sigma)) - \langle \sigma', x_t(\sigma) \rangle + \langle \sigma' - \sigma, x_t(\sigma') \rangle + 2\alpha \tag{31}$$

$$= \sum_{i=1}^{t-1} f_i(x_t(\sigma)) + m_t(x_t(\sigma)) - \langle \sigma, x_t(\sigma) \rangle + \langle \sigma' - \sigma, x_t(\sigma') - x_t(\sigma) \rangle + 2\alpha \tag{32}$$

Here, Equation (30) comes from the fact that $x_t(\sigma)$ is an approximate minimizer for the loss function, and Equation (31) comes from the fact that $x_t(\sigma')$ minimizes the loss function with noise set to $\sigma'$ by definition. Combining Equation (29) and Equation (32), we have that

$$0 \leq \langle \sigma' - \sigma, x_t(\sigma') - x_t(\sigma) \rangle + 2\alpha$$

$$\leq c(x_{(t,i)}(\sigma') - x_{(t,i)}(\sigma)) + 2\alpha$$

We, therefore, get that $x_{(t,i)}(\sigma') \geq x_{(t,i)}(\sigma) - \frac{2\alpha}{c}$.

$\square$

Moreover, we have that the difference between predictions made by our algorithm at sequential timesteps is close.

**Lemma D.3.** *If $\|x_t(\sigma) - x_{t+1}(\sigma)\|_1 \leq 10d|x_{t,i}(\sigma) - x_{t+1,i}(\sigma)|$ and $\sigma' = 100Lde_i + \sigma$, we have that*

$$\min\left(x_{t,i}(\sigma'), x_{t+1,i}(\sigma')\right) \geq \max(x_{t,i}(\sigma), x_{t+1,i}(\sigma)) - \frac{1}{10}|x_{t,i}(\sigma) - x_{t+1,i}(\sigma')| - \frac{3\alpha}{100Ld}$$

*Proof.* We have that

$$\sum_{i=1}^{t-1} f_i(x_t(\sigma)) - \langle \sigma, x_t(\sigma) \rangle + f_t(x_t(\sigma)) + m_{t+1}(x_t(\sigma))$$

$$\leq \sum_{i=1}^{t-1} f_i(x_{t+1}(\sigma)) - \langle \sigma, x_{t+1}(\sigma) \rangle + f_t(x_{t+1}(\sigma)) + +m_{t+1}(x_t(\sigma)) + \alpha \tag{33}$$

$$\leq \sum_{i=1}^{t-1} f_i(x_{t+1}(\sigma)) - \langle \sigma, x_{t+1}(\sigma) \rangle + f_t(x_{t+1}(\sigma)) + m_{t+1}(x_{t+1}(\sigma)) \tag{34}$$

$$+ L\|x_{t+1}(\sigma) - x_t(\sigma)\|_1 + \alpha$$

Here, we have Equation (33) from the approximate optimality of $x_t(\sigma)$ and Equation (34) from the smoothness of the optimistic function. Moreover, from the opposite direction, we have that

$$\sum_{i=1}^{t-1} f_i(x_t(\sigma)) - \langle \sigma, x_t(\sigma) \rangle + f_t(x_t(\sigma)) + m_{t+1}(x_t(\sigma))$$

$$= \sum_{i=1}^{t-1} f_i(x_t(\sigma)) - \langle \sigma', x_t(\sigma) \rangle + \langle \sigma' - \sigma, x_t(\sigma) \rangle + f_t(x_t(\sigma)) + m_{t+1}(x_t(\sigma))$$

$$\geq \sum_{i=1}^{t-1} f_i(x_t(\sigma')) - \langle \sigma', x_{t+1}(\sigma') \rangle + \langle \sigma' - \sigma, x_t(\sigma) \rangle + f_t(x_{t+1}(\sigma')) \tag{35}$$

$$+ m_{t+1}(x_{t+1}(\sigma')) - \alpha$$

$$= \sum_{i=1}^{t-1} f_i(x_t(\sigma')) - \langle \sigma, x_{t+1}(\sigma') \rangle + \langle \sigma' - \sigma, x_t(\sigma) - x_{t+1}(\sigma') \rangle + f_t(x_{t+1}(\sigma'))$$

$$+ m_{t+1}(x_{t+1}(\sigma')) - \alpha$$

$$\geq \sum_{i=1}^{t-1} f_i(x_t(\sigma)) - \langle \sigma, x_{t+1}(\sigma) \rangle + \langle \sigma' - \sigma, x_t(\sigma) - x_{t+1}(\sigma') \rangle + f_t(x_{t+1}(\sigma)) \tag{36}$$

$$+ m_{t+1}(x_{t+1}(\sigma)) - 2\alpha$$

Here, Equation (35) from the approximate optimality of $x_t(\sigma')$ and Equation (36) from the approximate optimality of $x_t(\sigma)$. From these and our original assumption, we have that,

$$10Ld\|x_{t+1,i}(\sigma) - x_{t,i}(\sigma)\|_1 + \alpha \geq 100Ld(x_{t,i}(\sigma) - x_{t+1,i}(\sigma')) - 2\alpha.$$

Using a similar argument, we have that

$$10Ld\|x_{t+1,i}(\sigma) - x_{t,i}(\sigma)\|_1 + \alpha \geq 100Ld(x_{t+1,i}(\sigma) - x_{t,i}(\sigma')) - 2\alpha.$$

Moreover, from Lemma D.2,we know that $x_{t+1,i}(\sigma') - x_{t+1,i}(\sigma) \geq \frac{-3\alpha}{100Ld}$ and $x_{t,i}(\sigma') - x_{t,i}(\sigma) \geq \frac{-3\alpha}{100Ld}$. Combining these, we have our claim. $\square$

We can now finally prove our claim. This proof is very similar to the proof of Theorem 1 in Suggala & Netrapalli (2019).

*Proof.* We note that we can decompose the $\ell_1$ norm in $\mathbb{E}\left[\|\mathbf{x}_t(\sigma) - \mathbf{x}_{t+1}(\sigma)\|_1\right]$ as

$$\mathbb{E}\left[\|\mathbf{x}_t(\sigma) - \mathbf{x}_{t+1}(\sigma)\|_1\right] = \sum_{i=1}^{d} \mathbb{E}\left[|\mathbf{x}_{t,i}(\sigma) - \mathbf{x}_{t+1,i}(\sigma)|\right].$$

To bound $\mathbb{E}\left[\|\mathbf{x}_t(\sigma) - \mathbf{x}_{t+1}(\sigma)\|_1\right]$ we derive an upper bound for each dimension $\mathbb{E}\left[|\mathbf{x}_{t,i}(\sigma) - \mathbf{x}_{t+1,i}(\sigma)|\right], \forall i \in [d]$. For any $i \in [d]$, define $\mathbb{E}_{-i}\left[|\mathbf{x}_{t,i}(\sigma) - \mathbf{x}_{t+1,i}(\sigma)|\right]$ as

$$\mathbb{E}_{-i}\left[|\mathbf{x}_{t,i}(\sigma) - \mathbf{x}_{t+1,i}(\sigma)|\right] = \mathbb{E}\left[|\mathbf{x}_{t,i}(\sigma) - \mathbf{x}_{t+1,i}(\sigma)| \mid \{\sigma_j\}_{j \neq i}\right]$$

where $\sigma_j$ is the $j^{\text{th}}$ coordinate of $\sigma$. Intuitively, we are computing in expectation of the noise of a single dimension while holding the other dimensions' noise constant. Let $\mathbf{x}_{\max,i}(\sigma) = \max\left(\mathbf{x}_{t,i}(\sigma), \mathbf{x}_{t+1,i}(\sigma)\right)$ and $\mathbf{x}_{\min,i}(\sigma) = \min\left(\mathbf{x}_{t,i}(\sigma), \mathbf{x}_{t+1,i}(\sigma)\right)$. Then, by definition, we have that

$$\mathbb{E}_{-i}\left[|\mathbf{x}_{t,i}(\sigma) - \mathbf{x}_{t+1,i}(\sigma)|\right] = \mathbb{E}_{-i}\left[\mathbf{x}_{\max,i}(\sigma)\right] - \mathbb{E}_{-i}\left[\mathbf{x}_{\min,i}(\sigma)\right].$$

Define event $\mathcal{E}$ as

$$\mathcal{E} = \left\{\sigma : \|\mathbf{x}_t(\sigma) - \mathbf{x}_{t+1}(\sigma)\|_1 \leq 10d \cdot |\mathbf{x}_{t,i}(\sigma) - \mathbf{x}_{t+1,i}(\sigma)|\right\}$$

For notational ease, let $\mathbf{P} = \exp(-100\eta L d)$ be a constant. Consider the following

$$
\begin{aligned}
\mathbb{E}_{-i}\left[\mathbf{x}_{\min,i}(\sigma)\right] = {} & \mathbb{P}\left(\sigma_i < 100Ld\right)\mathbb{E}_{-i}\left[\mathbf{x}_{\min,i}(\sigma) \mid \sigma_i < 100Ld\right] \\
& + \mathbb{P}\left(\sigma_i \geq 100Ld\right)\mathbb{E}_{-i}\left[\mathbf{x}_{\min,i}(\sigma) \mid \sigma_i \geq 100Ld\right] \\
\geq {} & (1 - \mathbf{P})\left(\mathbb{E}_{-i}\left[\mathbf{x}_{\max,i}(\sigma)\right] - D\right) && (37) \\
& + \mathbf{P}\mathbb{E}_{-i}\left[\mathbf{x}_{\min,i}\left(\sigma + 100Ld\mathbf{e}_i\right)\right] && (38)
\end{aligned}
$$

where Equation (37) follows from the fact that the domain of $i^{\text{th}}$ coordinate lies within some interval of length $D$ and since $\mathbb{E}_{-i}\left[\mathbf{x}_{\min,i}(\sigma) \mid \sigma_i < 100Ld\right]$ and $\mathbb{E}_{-i}\left[\mathbf{x}_{\max,i}(\sigma)\right]$ are points in this interval, their difference is bounded by $D$. We can further lower bound $\mathbb{E}_{-i}\left[\mathbf{x}_{\min,i}(\sigma)\right]$ as follows

$$
\begin{aligned}
\mathbb{E}_{-i}\left[\mathbf{x}_{\min,i}(\sigma)\right] \geq {} & (1 - \mathbf{P})\left(\mathbb{E}_{-i}\left[\mathbf{x}_{\max,i}(\sigma)\right] - D\right) \\
& + \mathbf{P}\mathbb{P}_{-i}(\mathcal{E})\mathbb{E}_{-i}\left[\mathbf{x}_{\min,i}\left(\sigma + 100Ld\mathbf{e}_i\right) \mid \mathcal{E}\right] \\
& + \mathbf{P}\mathbb{P}_{-i}\left(\mathcal{E}^c\right)\mathbb{E}_{-i}\left[\mathbf{x}_{\min,i}\left(\sigma + 100Ld\mathbf{e}_i\right) \mid \mathcal{E}^c\right]
\end{aligned}
$$

where $\mathbb{P}_{-i}(\mathcal{E})$ is defined as $\mathbb{P}_{-i}(\mathcal{E}) := \mathbb{P}\left(\mathcal{E} \mid \{\sigma_j\}_{j \neq i}\right)$. We now use the monotonicity properties proved in Lemma D.2 and Lemma D.3 to further lower bound $\mathbb{E}_{-i}\left[\mathbf{x}_{\min,i}(\sigma)\right]$. Then

$$
\begin{aligned}
\mathbb{E}_{-i}\left[\mathbf{x}_{min,i}(\sigma)\right] \geq {} & (1 - \mathbf{P})\left(\mathbb{E}_{-i}\left[\mathbf{x}_{\max,i}(\sigma)\right] - D\right) && (39) \\
& + \mathbf{P}\mathbb{P}_{-i}(\mathcal{E})\mathbb{E}_{-i}\left[\mathbf{x}_{\max,i}(\sigma) - \frac{1}{10}|\mathbf{x}_{t,i}(\sigma) - \mathbf{x}_{t+1,i}(\sigma)| - \frac{3\alpha}{100Ld} \mid \mathcal{E}\right] \\
& + \mathbf{P}\mathbb{P}_{-i}\left(\mathcal{E}^c\right)\mathbb{E}_{-i}\left[\mathbf{x}_{\min,i}(\sigma) - \frac{2\alpha}{100Ld} \mid \mathcal{E}^c\right] \\
\geq {} & (1 - \mathbf{P})\left(\mathbb{E}_{-i}\left[\mathbf{x}_{\max,i}(\sigma)\right] - D\right) && (40) \\
& + \mathbf{P}\mathbb{P}_{-i}(\mathcal{E})\mathbb{E}_{-i}\left[\mathbf{x}_{\max,i}(\sigma) - \frac{1}{10}|\mathbf{x}_{t,i}(\sigma) - \mathbf{x}_{t+1,i}(\sigma)| - \frac{3\alpha}{100Ld} \mid \mathcal{E}\right] \\
& + \mathbf{P}\mathbb{P}_{-i}\left(\mathcal{E}^c\right)\mathbb{E}_{-i}\left[\mathbf{x}_{\max,i}(\sigma) - \frac{1}{10d}\|\mathbf{x}_t(\sigma) - \mathbf{x}_{t+1}(\sigma)\|_1 - \frac{2\alpha}{100Ld} \mid \mathcal{E}^c\right]
\end{aligned}
$$

where Equation (39) follows from Lemma D.2 and Lemma D.3, Equation (40) follows from the definition of $\mathcal{E}^c$. Rearranging the terms in the RHS and using $\mathbb{P}_{-i}(\mathcal{E}) \leq 1$ gives us

$$
\begin{aligned}
\mathbb{E}_{-i}\left[\mathbf{x}_{\min,i}(\sigma)\right] \geq &(1-\mathbf{P})\left(\mathbb{E}_{-i}\left[\mathbf{x}_{\max,i}(\sigma)\right] - D\right) \\
&+ \mathbf{P}\mathbb{E}_{-i}\left[\mathbf{x}_{\max,i}(\sigma) - \frac{3\alpha}{100Ld}\right] \\
&- \mathbf{P}\mathbb{E}_{-i}\left[\frac{1}{10}\left|\mathbf{x}_{t,i}(\sigma) - \mathbf{x}_{t+1,i}(\sigma)\right| + \frac{1}{10d}\left\|\mathbf{x}_t(\sigma) - \mathbf{x}_{t+1}(\sigma)\right\|_1\right] \\
\geq &\mathbb{E}_{-i}\left[\mathbf{x}_{\max,i}(\sigma)\right] - 100\eta LdD - \frac{3\alpha}{100Ld} \\
&- \mathbb{E}_{-i}\left[\frac{1}{10}\left|\mathbf{x}_{t,i}(\sigma) - \mathbf{x}_{t+1,i}(\sigma)\right| + \frac{1}{10d}\left\|\mathbf{x}_t(\sigma) - \mathbf{x}_{t+1}(\sigma)\right\|_1\right]
\end{aligned}
\tag{41}
$$

where Equation (41) uses the the fact that $\exp(x) \geq 1 + x$. Rearranging the terms in the last inequality gives us

$$
\mathbb{E}_{-i}\left[\left|\mathbf{x}_{t,i}(\sigma) - \mathbf{x}_{t+1,i}(\sigma)\right|\right] \leq \frac{1}{9d}\mathbb{E}_{-i}\left[\left\|\mathbf{x}_t(\sigma) - \mathbf{x}_{t+1}(\sigma)\right\|_1\right] + \frac{1000}{9}\eta LdD + \frac{\mathbb{E}_{-i}[\alpha]}{30Ld}.
\tag{42}
$$

Since Equation (42) holds for any $\{\sigma_j\}_{j\neq i}$, we get the following bound on the unconditioned expectation

$$
\mathbb{E}\left[\left|\mathbf{x}_{t,i}(\sigma) - \mathbf{x}_{t+1,i}(\sigma)\right|\right] \leq \frac{1}{9d}\mathbb{E}\left[\left\|\mathbf{x}_t(\sigma) - \mathbf{x}_{t+1}(\sigma)\right\|_1\right] + \frac{1000}{9}\eta LdD + \frac{\mathbb{E}[\alpha]}{30Ld}.
\tag{43}
$$

Substituting Equation (43) with the above yields the following bound on the stability of predictions of FTPL+

$$
\mathbb{E}\left[\left\|\mathbf{x}_t(\sigma) - \mathbf{x}_{t+1}(\sigma)\right\|_1\right] \leq 125\eta Ld^2D + \frac{\alpha}{20L}
$$

Utilizing Appendix D.1 gives us the required bound on regret. □

## D.2 Proofs for Strong Gradient Dominance

We will first prove what the gradient of the value function is with respect to $\theta$ in this setting.

**Lemma 6.2.** *We have $g(W, \pi) = V_W^\pi(\mu) - \|\pi\|_2^2$ is $\mathcal{K}_\pi, \frac{T}{2}$ strongly gradient-dominated on set $\mathcal{T}$*

*Proof.* By the Performance Difference Lemma,

$$\sum_t V_{W_t}^{\pi^*} - \sum_t V_{W_t}^{\pi} - \langle \sigma, \pi - \pi^* \rangle + \|\pi^*\|_2^2 - \|\pi\|_2^2$$

$$= \frac{1}{1-\gamma} \sum_t \sum_{s,a} d_\mu^{\pi^*}(s)\pi^*(a,s)A^\pi(s,a) - \langle \sigma, \pi - \pi^* \rangle + \|\pi^*\|_2^2 - \|\pi\|_2^2 \tag{44}$$

$$\leq -\min_{\bar{\pi}}\left[ \frac{-1}{1-\gamma}\sum_t\sum_{s,a} d_\mu^{\pi^*}(s)\bar{\pi}(s,a)A^\pi(s,a) - \langle \sigma, \pi^* - \pi \rangle - \|\pi^*\|_2^2 + \|\pi\|_2^2 \right]$$

$$\leq -\left\| \frac{d_\mu^{\pi^*}}{d_\mu^{\pi}} \right\|_\infty \min_{\bar{\pi}}\left[ \frac{-1}{1-\gamma}\sum_t\sum_{s,a} d_\mu^{\pi}(s)\bar{\pi}(s,a)A^\pi(s,a) - \langle \sigma, \bar{\pi} - \pi \rangle - \|\bar{\pi}\|_2^2 + \|\pi\|_2^2 \right]$$

$$= -\left\| \frac{d_\mu^{\pi^*}}{d_\mu^{\pi}} \right\|_\infty \min_{\bar{\pi}}\left[ \frac{-1}{1-\gamma}\sum_t\sum_{s,a} d_\mu^{\pi}(s)(\bar{\pi}(s,a) - \pi(a,s))A^\pi(s,a) - \langle \sigma, \bar{\pi} - \pi \rangle \right. \tag{45}$$

$$\left. - \|\bar{\pi}\|_2^2 + \|\pi\|_2^2 \right]$$

$$= -\left\| \frac{d_\mu^{\pi^*}}{d_\mu^{\pi}} \right\|_\infty \min_{\bar{\pi}}\left[ \frac{-1}{1-\gamma}\sum_t\sum_{s,a} d_\mu^{\pi}(s)(\bar{\pi}(s,a) - \pi(a,s))Q^\pi(s,a) - \langle \sigma, \bar{\pi} - \pi \rangle \right. \tag{46}$$

$$\left. - \|\bar{\pi}\|_2^2 + \|\pi\|_2^2 \right]$$

$$= -\left\| \frac{d_\mu^{\pi^*}}{d_\mu^{\pi}} \right\|_\infty \min_{\bar{\pi}}\left[ \frac{-1}{1-\gamma}\sum_t\sum_{s,a} d_\mu^{\pi}(s)(\bar{\pi}(s,a) - \pi(a,s))Q^\pi(s,a) - \langle \sigma, \bar{\pi} - \pi \rangle \right. \tag{47}$$

$$\left. - \langle 2\pi, \bar{\pi} - \pi \rangle - \|\bar{\pi} - \pi\|_2^2 \right]$$

$$\leq \frac{-1}{1-\gamma}\left\| \frac{d_\mu^{\pi^*}}{\mu} \right\|_\infty \min_{\bar{\pi}}\left[ (\bar{\pi} - \pi)^\top \nabla_\pi\left( -\sum_t V_{W_t}^\pi - \langle \sigma, \pi \rangle - T\|\pi\|_2^2 \right) - \frac{T}{2}\|\bar{\pi} - \pi\|_2^2 \right] \tag{48}$$

Here, Equation (44) comes from the Performance Difference Lemma, Equation (45) comes from the fact that $\sum_a \pi(a,s)A^\pi(s,a) = 0$, Equation (46) comes from the definition of the Advantage Function for the $\pi$-player, and Equation (48) comes from the both the Policy Gradient Theorem and the fact that $d_\mu^\pi(s) \geq (1-\gamma)\mu(s)$. Moreover, Equation (47) comes from the following logic

$$-\|\bar{\pi} - \pi\|_2^2 = -\left[ \bar{\pi}^\top\bar{\pi} - 2\bar{\pi}^\top\pi + \pi^\top\pi \right]$$

$$= -\left[ \bar{\pi}^\top\bar{\pi} - 2\bar{\pi}^\top\pi + 2\pi^\top\pi - \pi^\top\pi \right]$$

$$= -\left[ \|\bar{\pi}\|_2^2 - \|\pi\|_2^2 + 2\langle \pi - \bar{\pi}, \pi \rangle \right]$$

$$= -\|\bar{\pi}\|_2^2 + \|\pi\|_2^2 + 2\langle \bar{\pi} - \pi, \pi \rangle$$

From this, we have that the value function done in this manner is strongly gradient dominated with constant $\frac{T}{2}$. □

### D.3 Proof of Theorem 5.1

**Theorem 5.1.** *Assume that the set of possible transition matrices $\mathcal{W}$ is convex. Let $L_\pi$ be the smoothness constant of $h_t$ with respect to the $\ell_1$ norm, $D_\pi = \max_{\pi,\pi'\in\mathcal{T}}\|\pi - \pi'\|_2$, and $d_\pi$ be the dimension of the input for the $\pi$-player. Let $OL^\pi$ use FTPL and $OL^W$ use Best-Response. Under direct parameterization, when*

$\eta = \frac{1}{L_\pi \sqrt{T d_\pi}}$ the robustness of the set of trained algorithms is for any $\bar{\pi}$

$$\min_{W^* \in \mathcal{W}} \sum_{t=0}^{T} V_{W^*}^{\bar{\pi}}(\mu) - \mathbb{E}\left[\min_{W^* \in \mathcal{W}} \sum_{t=0}^{T} V_{W^*}^{\pi_t}(\mu)\right] \leq \frac{2 d_\pi^{\frac{3}{2}} D_\pi L_\pi}{\sqrt{T}} + c_\pi(T_\mathcal{O}, \mathcal{K}_\pi) + c_W(T_\mathcal{O}, \mathcal{K}_W).$$

*Proof.* From Theorem 4.1, we have that

$$\min_{W^*} \sum_{t=0}^{T} V_{W^*}^{\bar{\pi}}(\mu) - \min_{W^*} \sum_{t=0}^{T} V_{W^*}^{\pi_t}(\mu) \leq \text{Reg}_W + \text{Reg}_\pi.$$

When the $\pi$-player is using FTPL, its regret is bounded according to

$$\mathbb{E}(\text{Reg}_\pi) \leq \mathcal{O}\left(\eta d_\pi^2 D_\pi L_\pi^2 + \frac{d_\pi D_\pi}{\eta T} + \alpha\right).$$

Setting $\eta = \frac{1}{L_\pi \sqrt{T d_\pi}}$ to minimize this, we have

$$\mathbb{E}(\text{Reg}_\pi) \leq \mathcal{O}\left(\frac{2 d_\pi^{\frac{3}{2}} D_\pi L_\pi}{\sqrt{T}} + \alpha\right).$$

Moreover, $\alpha$ is the Oracle Error term. Therefore, given that we have gradient dominance, using Projected Gradient Descent yields from Lemma 5.1

$$\mathbb{E}(\text{Reg}_\pi) \leq \mathcal{O}\left(\frac{2 d_\pi^{\frac{3}{2}} D_\pi L_\pi}{\sqrt{T}} + c_\pi(T_\mathcal{O}, \mathcal{K}_\pi)\right).$$

Given the $W$-player employs Best-Response, we have that the regret of the $W$-player is bounded by the following by Lemma 4.1

$$\mathbb{E}(\text{Reg}_W) \leq \alpha$$

where $\alpha$ is the optimization error. Given that we have gradient dominance properties for the $W$-player as well, we have that using the Projected Gradient Descent for

$$\mathbb{E}(\text{Reg}_W) \leq c_W(T_\mathcal{O}, \mathcal{K}_W).$$

Adding these two inequalities together gets our final result. □

### D.4   Proof of Theorem 6.1

**Theorem 6.1.** *Assume that the set of possible transition matrices $\mathcal{W}$ is convex and we are in the direct parameterization setting. Let $OL^\pi$ use OFTPL and $OL^W$ use FTPL+. Given that Condition 6.1 holds, we have that the robustness of the algorithm is for any $\bar{\pi}$ when setting $\eta = \sqrt{\frac{20 L_\pi (d_w D_W + d_\pi D_\pi)^2}{d_\pi} D_\pi \tilde{L}_2 \alpha T}$,*

$$\min_{W^* \in \mathcal{W}} \sum_{t=0}^{T} V_{W^*}^{\bar{\pi}}(\mu) - \mathbb{E}\left[\min_{W^* \in \mathcal{W}} \sum_{t=0}^{T} V_{W^*}^{\pi_t}(\mu)\right] \leq \mathcal{O}\left(\left[\frac{d_\pi \tilde{L} \sqrt{D_\pi (d_w D_w + d_\pi D_\pi)}}{\sqrt{5 L_\pi T c_\pi(T_\mathcal{O}, K_\pi)}} + 1\right] c_\pi(T_\mathcal{O}, K_\pi) + c_W(T_\mathcal{O}, \mathcal{K}_W)\right).$$

*Proof.* From Theorem 4.1, we have that

$$\min_{W^*} \sum_{t=0}^{T} V_{W^*}^{\bar{\pi}}(\mu) - \min_{W^*} \sum_{t=0}^{T} V_{W^*}^{\pi_t}(\mu) \leq \text{Reg}_W + \text{Reg}_\pi.$$

Given the $\pi$-player employs OFTPL, we have from Lemma C.1 that the regret is upper bounded by

$$\text{Reg}_\pi \leq O\left(\eta d_\pi^2 D_\pi \sum_{t=1}^{T} \frac{L_t^2}{T} + \frac{d_\pi D_\pi}{\eta T} + \alpha\right).$$

However, we have that

$$\mathbb{E}(L_t^2) = \tilde{L}^2 \|W_t - W_{t-1}\|_1$$

$$\leq \tilde{L}^2 \left( 125\eta L_\pi d_\pi^2 D_\pi + \frac{\alpha}{20 L_\pi} \right)$$

Here, the last inequality comes from the fact that the $W$-player uses FTPL+, and the decisions made by FTPL+ are stable from Lemma D.1. Using this above, we have that

$$\text{Reg}_\pi \leq O\left( \eta d_\pi^2 D_\pi \tilde{L}^2 \left( 125\eta L_\pi d_\pi^2 D_\pi + \frac{\alpha}{20 L_\pi} \right) + \frac{d_\pi D_\pi}{\eta T} + \alpha \right).$$

Since our $\pi$-player enjoys gradient dominance for its loss function, we have from Lemma 5.1 that

$$\alpha \leq c_\pi \left( T_\mathcal{O}, \mathcal{K}_\pi \right).$$

Moreover, given the $W$-player is employing FTPL+, from Lemma 4.2, we have that regret of the $W$-player is bounded by

$$\text{Reg}_W \leq \mathcal{O}\left( \frac{d_W D_W}{\eta T} + \alpha \right).$$

In this setting, the $W$-player still enjoys gradient dominance properties, so using Projected Gradient Descent has

$$\alpha \leq c_W \left( T_\mathcal{O}, \mathcal{K}_W \right).$$

Adding these together yields

$$\text{Reg}_\pi + \text{Reg}_W \leq \mathcal{O}(\eta d_\pi^2 D_\pi \tilde{L}^2 \left( 125\eta L_\pi d_\pi^2 D_\pi + \frac{\alpha}{20 L_\pi} \right) + \frac{d_W D_W + d_\pi D_\pi}{\eta T} +$$

$$c_\pi \left( T_\mathcal{O}, \mathcal{K}_\pi \right) + c_W \left( T_\mathcal{O}, \mathcal{K}_W \right)).$$

Setting $\eta = \sqrt{\frac{20 L_\pi (d_w D_W + d_\pi D_\pi)^2}{d_\pi} D_\pi \tilde{L}_2 \alpha T}$, we have that

$$\text{Reg}_\pi + \text{Reg}_W \leq \mathcal{O}\left( \left[ \frac{d_\pi \tilde{L} \sqrt{D_\pi (d_w D_w + d_\pi D_\pi)}}{\sqrt{5 L_\pi T c_\pi (T_\mathcal{O}, K_\pi)}} + 1 \right] c_\pi (T_\mathcal{O}, K_\pi) + c_W \left( T_\mathcal{O}, \mathcal{K}_W \right) \right).$$

$\square$

### D.5 Proof for Theorem 6.2

**Theorem 6.2.** *We are in the direct parameterization setting where the objective function is the regularized Value Function. Assume that the set of possible transition matrices $\mathcal{W}$ is convex. Let $OL^\pi$ use FTPL and $OL^W$ use Best-Response. Under direct parameterization, given that Condition 6.2 holds and $\eta = \frac{1}{L_\pi \sqrt{T d_\pi}}$, we have that the robustness of Algorithm 1 is for any $\bar{\pi}$*

$$\min_{W^* \in \mathcal{W}} \sum_{t=0}^T V_{W^*}^{\bar{\pi}}(\mu) - \|\bar{\pi}\|^2 - \min_{W^* \in \mathcal{W}} \sum_{t=0}^T V_{W^*}^{\pi_t}(\mu) 0 \|\pi_t\|^2 \leq \frac{2 d_\pi^{\frac{3}{2}} D_\pi L_\pi}{\sqrt{T}} + c_\pi^s \left( T_\mathcal{O}, \mathcal{K}_\pi, \frac{1}{2} \right) + c_W (T_\mathcal{O}, \mathcal{K}_W).$$

*Proof.* From Theorem 4.1, we have that

$$\min_{W^*} \sum_{t=0}^T V_{W^*}^{\bar{\pi}}(\mu) - \min_{W^*} \sum_{t=0}^T V_{W^*}^{\pi_t}(\mu) \leq \text{Reg}_W + \text{Reg}_\pi.$$

When the $\pi$-player is using FTPL, its regret is bounded according to

$$\mathbb{E}(\text{Reg}_\pi) \leq \mathcal{O}\left( \eta d_\pi^2 D_\pi L_\pi^2 + \frac{d_\pi D_\pi}{\eta T} + \alpha \right).$$

Setting $\eta = \frac{1}{L\sqrt{Td}}$ to minimize this, we have

$$\mathbb{E}(\text{Reg}_\pi) \leq \mathcal{O}\left(\frac{2d_\pi^{\frac{3}{2}} D_\pi L_\pi}{\sqrt{T}} + \alpha\right).$$

Moreover, $\alpha$ is the Oracle Error term. Therefore, given that we have strong gradient dominance, using Projected Gradient Descent yields from Lemma 6.1

$$\mathbb{E}(\text{Reg}_\pi) \leq \mathcal{O}\left(\frac{2d_\pi^{\frac{3}{2}} D_\pi L_\pi}{\sqrt{T}} + +c_\pi^s\left(T_\mathcal{O}, \mathcal{K}_\pi, \frac{1}{2}\right)\right).$$

Given the $W$-player employs Best-Response, we have that the regret of the $W$-player is bounded by the following by Lemma 4.1

$$\mathbb{E}(\text{Reg}_W) \leq \alpha$$

where $\alpha$ is the optimization error. Given that we have gradient dominance properties for the $W$-player as well, we have that using the Projected Gradient Descent for

$$\mathbb{E}(\text{Reg}_W) \leq c_W\left(T_\mathcal{O}, \mathcal{K}_W\right).$$

Adding these two inequalities together gets our final result. $\qquad\square$

