# OpenReview forum: "Solving Robust MDPs through No-Regret Dynamics"
_TMLR — Accepted by TMLR_

### Review · Reviewer_E21q · 2024-04-16

**Summary Of Contributions:**

This paper studies the problem of solving robust MDPs. The authors propose to use a No-Regret Dynamics framework that iterates between policy learning and worst-case transition approximation. This two-player framework doesn't require stringent assumptions about the uncertainty set except convexity. A toolbox of non-convex online learning algorithms for both players is provided. By leveraging the gradient dominance property of robust MDPs, the framework can use Projected Gradient Descent as an optimization oracle for the online algorithms. With that, the proposed algorithm can yields an $\mathcal{O}(T^{-\frac{1}{2}})$ convergence rate. In addition, the authors also discuss achieving faster convergence rates under stronger assumptions (e.g., smoothness).

**Audience:**

Yes

**Claims And Evidence:**

Yes

**Requested Changes:**

See above

**Strengths And Weaknesses:**

Strength:
* The writing is clear and easy to follow.
* The paper makes important contributions towards solving Robust MDPs under little structure assumptions. The proposed framework is general, versatile, and suitable for various problem settings.

Comment and questions:
* Including some discussions on implementing a practical algorithm based on the theoretical framework would further strengthen the paper.
* Why FTPL and OFTPL are suitable for the policy player?
* What is $d_w$ in Theorem 6.1?
* The framework has a general loss $g(W,\pi)$. Can you give more examples? Can it be used in, say, pure exploration?

Minor changes:
* Including descriptions for the GridWorld MDP (for completeness) would be better for reading.
* Bottom line on Page 1, "Several works": Please add the references.

---

> ### Author Response · Authors · 2024-05-05
> **Response**
>
> We appreciate this reviewer's thoughtful review! Thank you for your time and insights! We have responded to each of your concerns. Please let us know if you have any follow-up questions.
>
> ## Weaknesses
> >W1.  Including some discussions on implementing a practical algorithm based on the theoretical framework would further strengthen the paper.
>
> A1. Thank you for the helpful advice! In Section 6.1 of the revised version, we provide some advice on how to implement our algorithms.
>
>
> ## Questions
> > Q1. Why FTPL and OFTPL are suitable for the policy player?
>
> A1. Since they do not need to see the incoming loss transition function, FTPL and OFTPL are suitable choices for the policy player.
>
> &nbsp;
>
>
> > Q2. What is $d_w$ in Theorem 6.1?
>
> A2. $d_W$ is the dimension of the parameterization of the environment.
>
> &nbsp;
>
>
> > Q3. The framework has a general loss $g(W, \pi)$. Can you give more examples? Can it be used in, say, pure exploration?
>
> A3. In Section 7.3, we explore where the objective function is the Value Function with regularization $g(W, \pi) = V_W^{\pi}(\mu) - \| \pi \|_2^2$. For our framework to apply, the objective function's main requirement is to exhibit gradient dominance for both the policy and environment players. Bhandari and Russo (2022) explore several objective functions where gradient dominance holds for the policy player.
>
> &nbsp;
>
>
> ## Minor changes:
> > M1. Including descriptions for the GridWorld MDP (for completeness) would be better for reading.
>
> A1. Thank you for the suggestion. We have added a description in Section 7.
>
> &nbsp;
>
>
> > M2. Bottom line on Page 1, "Several works": Please add the references.
>
> A2. Thank you for the suggestion. We have added the references.
> ## References
> Jalaj Bhandari and Daniel Russo. Global optimality guarantees for policy gradient methods. 2022.

---

> > ### Comment · Reviewer_E21q · 2024-05-07
> > **Thank you**
> >
> > Thank the authors for replying to my comments. The response has addressed my questions.

---

### Review · Reviewer_ypfF · 2024-04-17

**Summary Of Contributions:**

The authors propose a framework for solving robust MDPs based on no-regret dynamics. Different algorithms are provided under this framework: Best-Response and Follow the Perturbed Leader Plus (FTPL+) are proposed. Further, under the nonconvexity of the setting, a usual necessity of these algorithms is the need for a global minimizer oracle. The show that for robust MDPs, the value function shows gradient-dominance, a sufficient condition to use projected gradient descent as an oracle. Convergence rates are provided for the algorithm of study.

**Audience:**

Yes

**Claims And Evidence:**

Yes

**Requested Changes:**

The aspects previously mentioned should be addressed.

**Strengths And Weaknesses:**

The paper aims to address an interesting and important problem through the lens of a unified framework. My main concert with the work lies on its formality and presentation.

- It is not clear what is the exact contribution of this work is, given that the proposed algorithms are already known. It looks like the application of these algorithms to the specific scenario of this work. The formalization of them into a framework for robust MDPs can be a valuable contribution, but it's presentation should be clarified.

- The use of notation is sometimes confusing, e.g., the double definition of the probability of the state distribution d^W and d^\pi? Why the introduction of a generic objective g(\pi,W), when the rest of the paper relies on the value function? Overall the paper lacks formalism.

- The lack of equation numbering and lack of referencing across the paper makes it sometimes difficult to follow. This is specially so in the appendices, where proofs are presented in a not completely formal way, which makes it difficult to check for correctness. E.g, Lemma 5.2 claims, "[...] the third inequality comes from [...]", instead of using numbering and in this case, there are no inequalities in the previous equation.

---

> ### Author Response · Authors · 2024-05-05
> **Response**
>
> We appreciate this reviewer's thoughtful review! Thank you for your time and insights! We have responded to each of your concerns. Please let us know if you have any follow-up questions.
>
> ## Weaknesses
>
> > W1. It is not clear what is the exact contribution of this work is, given that the proposed algorithms are already known. It looks like the application of these algorithms to the specific scenario of this work. The formalization of them into a framework for robust MDPs can be a valuable contribution, but it's presentation should be clarified.
>
> A1. We establish a framework to design algorithms for different Robust MDPs alongside a simple proof framework. While the constituent building blocks exist, we motivate how to pair them together to get a powerful algorithm for Robust MDPs. The results are algorithms with some of the strongest convergence rates for Robust MDPs without rectangularity assumptions on the uncertainty set.
>
> &nbsp;
>
> > W2. The use of notation is sometimes confusing, e.g., the double definition of the probability of the state distribution d^W and d^\pi? Why the introduction of a generic objective g(\pi,W), when the rest of the paper relies on the value function? Overall the paper lacks formalism.
>
> A2. We apologize for the confusion around our notation for the occupancy measure. Since the occupancy measure depends on the start state, policy, and environment, we chose the notation of $\pi$ implied in $d_s^W$ and $W$ implied in $d_s^{\pi}$ for simplicity. If you have a suggestion for changing this, please let us know! Moreover, we wanted to mention that we do not only consider one objective function. In Section 6.3, we give the example of where the objective function is in the Value Function under regularization, i.e., $g(W, \pi) = V_W^{\pi}(\mu) - \| \pi \|_2^2$. There are different formulations, such as pure exploration, that Reviewer E21q asked for.
>
> &nbsp;
>
> > W3. The lack of equation numbering and lack of referencing across the paper makes it sometimes difficult to follow. This is specially so in the appendices, where proofs are presented in a not completely formal way, which makes it difficult to check for correctness. E.g, Lemma 5.2 claims, "[...] the third inequality comes from [...]", instead of using numbering and in this case, there are no inequalities in the previous equation.
>
> A3. Thank you for the suggestion! In the revised version, we have added referencing and equation labels throughout the paper. We greatly appreciate the advice!

---

### Review · Reviewer_BhiD · 2024-04-26

**Summary Of Contributions:**

This paper studies designing an algorithm that can find environmentally robust policies efficiently and handle different model parameterizations. To solve this problem, this paper proposes a No-Regret Dynamics framework that utilizes policy gradient methods and iteratively approximates the worst case environment during training, avoiding assumptions on the uncertainty set. With a toolbox of nonconvex online learning algorithms, the authors demonstrate that the proposed framework can achieve fast convergence rates for many problem settings and relax assumptions on the uncertainty set of transitions.

**Audience:**

Yes

**Broader Impact Concerns:**

I do not have ethical concerns.

**Claims And Evidence:**

Yes

**Requested Changes:**

Please see the review on weaknesses above.

**Strengths And Weaknesses:**

**Strengths:**

1.	The studied problem, designing an algorithm that can find environmentally robust policies and handle different model parameterizations, is an interesting and important to the MDP literature.
2.	The authors design an algorithmic framework for generating algorithms to solve Robust MDPs based on No-Regret Dynamics. Under this framework, the authors further develop novel no-regret online learning algorithms. This framework works for various problem settings, including direct parameterization. Moreover, due to the explicit approximation of the worst-case environment during training, to guarantee convergence, this paper only needs the assumption that the uncertainty set of transitions is convex.
3.	This paper shows that when the objective function of the game is gradient-dominated for both players, one can use Projected Gradient Descent as the oracle. This paper provides tools for identifying the conditions under which the gradient-dominated property of an objective function is ensured. This paper proves that when the objective function is the Value Function in the direct parameterization setting, the proposed algorithm achieves an $O(T^{-1/2})$ convergence rate.
4.	This paper demonstrates that when the objective function is Lipschitz smooth or enjoys strong gradient-dominance with advanced online dynamics, faster convergence rates can be obtained.
5.	This paper is well-written and easy to follow.

**Weaknesses:**

1.	The experiments should be put in the main paper, instead of Appendix. It seems that the experiments only include the results of the proposed algorithm. It would enhance the experiment part if the authors can add more baselines, e.g., the adaptations of prior algorithms or ablated variations, and compare the empirical performance of the proposed algorithm and baselines.
2.	It would facilitate the readers’ understanding if the authors can give more discussions on the connection between the proposed algorithmic framework and results and existing algorithms and results in the no-regret and robust MDP literature.
3.	In addition to direct parameterization, softmax parameterization is also widely used in policy gradient methods for RL. Can the proposed algorithmic framework work for softmax parameterization? Can the authors give some comments on possible extensions to softmax parameterization?

Overall, I think this submission is a good paper, and would further improve if the authors can address the above comments. I tend to weakly accept.

---

> ### Author Response · Authors · 2024-05-05
> **Response**
>
> We appreciate this reviewer's thoughtful review! Thank you for your time and insights! We have responded to each of your concerns. Please let us know if you have any follow-up questions.
>
> ## Weaknesses
> > W1. The experiments should be put in the main paper, instead of Appendix. It seems that the experiments only include the results of the proposed algorithm. It would enhance the experiment part if the authors can add more baselines, e.g., the adaptations of prior algorithms or ablated variations, and compare the empirical performance of the proposed algorithm and baselines.
>
> A1. Thank you for the suggestions. We originally placed the experiments in the appendix as they are not the main contribution of this paper. We have moved the experiments to the main text and added the Double-Loop Robust Policy Gradient (DRPG) as a baseline from Wang et al. (2022b). The revised version also adds the new convergence charts in the experimental section and a result comparison between the our method and DRPG. We see that our method and DRPG get very similar convergence curves over many settings up to a difference of noise.
>
> &nbsp;
>
> > W2. It would facilitate the readers' understanding if the authors can give more discussions on the connection between the proposed algorithmic framework and results and existing algorithms and results in the no-regret and robust MDP literature.
>
> A2. Thank you for the helpful suggestion! In Section 5.5, highlighted in green in the revised version, we have added a further discussion connecting our algorithm to previous algorithms in the literature.
>
> &nbsp;
>
> > W3. In addition to direct parameterization, softmax parameterization is also widely used in policy gradient methods for RL. Can the proposed algorithmic framework work for softmax parameterization? Can the authors give some comments on possible extensions to softmax parameterization?
>
> A3. Thank you for the suggestion. Under certain conditions, our algorithm works for the Softmax Parameterization. The main question is whether we can characterize an approximate optimization oracle of the policy player's objective function under the Softmax Parameterization. For example,  Agarwal et al. (2021) established asymptotic global convergence of softmax PG methods for infinite-horizon $\gamma$-discounted tabular MDPs. Therefore, in this setting, Softmax Policy Gradient methods can function as the approximate optimization oracle for the policy player. Thus, our framework will work in this setting. Expanding to different parameterizations only requires finding an approximate optimization oracle.
>
>
>
> ## References
> Agarwal, A., Kakade, S. M., Lee, J. D., and Mahajan, G. (2021). On the theory of policy gradient methods: Optimality, approximation, and distribution shift. Journal of Machine Learning Research, 22(98):1–76.
>
> &nbsp;
>
> Qiuhao Wang, Chin Pang Ho, and Marek Petrik. On the convergence of policy gradient in robust mdps. arXiv preprint arXiv:2212.10439, 2022b

---

### Decision · Action_Editor_1ycn · 2024-05-28

**Recommendation:** Accept as is

**Comment:**

The reviewers felt that the revisions made by the authors during the review process satisfied their main questions and concerns.

**Audience:**

Yes, the reviewers agree that the paper offers insight regarding an important problem (robust RL).

**Claims And Evidence:**

Yes, the reviewers found the claims of the paper to be well-supported by proofs and experiments.